



# Meteorological and cloud conditions during the Arctic Ocean 2018 expedition

Jutta Vüllers[1], Peggy Achtert[2], Ian M. Brooks[1], Michael Tjernström[3], John Prytherch[3], Ryan Neely III[1]

[1]Institute for Climate and Atmospheric Science, School of Earth and Environment, University of Leeds, Leeds, LS2 9JT, United Kingdom
[2] Meteorological Observatory Hohenpeißenberg, German Weather Service, Hohenpeißenberg, 82383, Germany
[3]Department of Meteorology, Stockholm University, Stockholm, 10691, Sweden

*Correspondence to*: Jutta Vüllers (j.vuellers@leeds.ac.uk)

**Abstract.** The Arctic Ocean 2018 (AO2018) expedition took place in the central Arctic Ocean in August and September 2018. An extensive suite of instrumentation provided detailed measurements of surface water chemistry and biology, sea ice and ocean physical and biogeochemical properties, surface exchange processes, aerosols, clouds, and the state of the atmosphere.

The measurements provide important information on the coupling of the ocean and ice surface to the atmosphere and in particular to clouds. This paper provides: i) an overview of the synoptic-scale atmospheric conditions and its climatological anomaly to help interpret the process studies and put the detailed observations from AO2018 into a larger context, both spatially and temporally; ii) a statistical analysis of the thermodynamic and near-surface meteorological conditions, boundary layer, cloud, and fog characteristics; iii) a comparison of the results to observations from earlier Arctic Ocean expeditions, in

particular AOE96, SHEBA, AOE2001, ASCOS, ACSE, and AO2016, to provide an assessment of the representativeness of the measurements. The results show that near-surface conditions were broadly comparable to earlier experiments, however the thermodynamic vertical structure was quite different. An unusually high frequency of well-mixed boundary layers up to about 1 km depth occurred, and only a few cases of the "prototypical" Arctic summer single-layer stratocumulus deck were observed. Instead, an unexpectedly high amount of multiple cloud layers and mid-level clouds was present throughout the campaign.

These differences from previous studies are related to the high frequency of cyclonic activity in the central Arctic in 2018.

## 1 Introduction

The climate in the Arctic is changing rapidly (Richter-Menge et al., 2018). Arctic near-surface temperature has continuously increased over recent decades and the warming is two to three times larger than the global mean (Hartfiel et al., 2018; IPCC, 2018; Serreze and Barry, 2011). This phenomenon is commonly referred to as Arctic amplification. The past five years (2014-

2018) were the warmest in the record starting in 1900 (Osborne et al., 2018).

An obvious manifestation of the changing Arctic is the sea ice loss. A strong reduction in sea ice cover and thickness has been recorded over 40 years (Onarheim et al., 2018; Stroeve et al., 2012), and multiyear sea ice cover is shrinking (Richter-Menge et al., 2018). In 2018, less than 1% of the Arctic sea ice was more than 4 years old; a decline of 95% compared to 1985 (Osborne et al., 2018).

Even though there is consensus on these phenomena, the understanding of the underlying processes is limited (Wendisch et al., 2019). Multiple feedback processes contribute to the Arctic amplification, including surface albedo feedback (Taylor et al., 2013; Perovich et al., 2008), cloud feedbacks (Holland and Bitz, 2003; Liu et al., 2008, Taylor et al., 2013), and dynamic transport feedback (Graversen et al., 2008; Boeke and Taylor, 2016). The limited understanding is also reflected by the particularly large spread in climate model projections for the Arctic. The Coupled Model Intercomparison Project 3 (CMIP3)

(IPCC, 2007) and CMIP5 (IPCC, 2013) climate models agree on the warming trend in the Arctic, however, the model spread in surface temperature increase is much larger for the Arctic region than for other regions (Pithan and Mauritsen, 2014). This is mainly related to inadequate sub-grid-scale parameterisations, unable to represent the unique Arctic environment (Hodson et al., 2013; Vihma et al., 2014).

Cloud feedback processes in the Arctic are particularly challenging for models as there are notable differences to the more

commonly studied mid-latitudes and tropics. These differences are:

(i) The climatologically near-ubiquitous stratus clouds in summer. These clouds are often persistent mixed phase clouds (Shupe et al., 2011; Shupe, 2011), which are particularly challenging for models as they are in an unstable thermodynamic state. Several intimately coupled processes are involved in creating this resilient mixed phase cloud system: radiative cooling, turbulent mixing, ice and cloud droplet formation and growth, entrainment, and turbulent

surface fluxes (Morrison et al., 2012). These clouds modulate the surface energy budget considerably (e.g. Intrieri et al., 2002; Shupe and Intrieri, 2004). Relative to clear sky conditions these low-level clouds often have a warming effect on the surface, instead of a cooling effect as is the case for lower latitudes (Sedlar et al. 2011).

(ii) Very low aerosol concentrations, in particular cloud condensation nuclei (CCN) and ice nucleating particles (INP), whose sources are still unclear, and which are a significant controlling factor for cloud radiative properties (Birch et

al., 2012; Mauritsen et al., 2011; Prenni et al., 2007).

(iii) Humidity inversions across cloud tops, so that entrainment becomes a source of moisture to the boundary layer and hence helping to sustain the persistent stratus clouds against water losses from precipitation (Shupe et al., 2013).

As a result of these factors, the representation of Arctic clouds is challenging for models, and the influence of clouds on the energy budget is highly uncertain in climate projections. Hence, there is an urgent need to improve model parameterisations

which requires a better understanding of the physical processes involved; this process understanding can only be achieved from the analysis of direct, detailed, in-situ measurements. These are also necessary for testing new parameterisations. A number of field campaigns aimed at this challenge have been conducted in the Arctic over the last 25 years. The campaigns focused on different processes including air-ice-sea interactions, the surface heat and energy budget, aerosol-cloud interactions,



and cold-air outbreaks (Wendisch et al. 2019, and references therein). They were conducted in different parts of the Arctic and during different times of the year, though primarily in the Arctic summer and the beginning of the freeze up.

The Arctic Ocean 2018 (AO2018) campaign was conducted in roughly the same area and during a similar time of the year as four of the previous campaigns: the Arctic Ocean Expedition 1996 (AOE-96, Leck et al., 2001); the Arctic Ocean Experiment 2001 (AOE-2001, Tjernström et al., 2004a,b); the Arctic Summer Cloud Ocean Study (ASCOS) (Tjernström et al., 2014); and the Arctic Ocean 2016 (AO2016) expedition. During AO2018 extensive and coordinated atmospheric near-surface and remote sensing measurements of clouds, boundary layer properties, and aerosol particles were conducted. As the atmosphere is highly variable and synoptic conditions vary from year to year, it is important to compare the newly gained results to those from previous campaigns to gauge how representative the measurements are. This paper summarizes the meteorological conditions during AO2018 and puts the measurements into the contexts of both the synoptic setting and of the measurements from previous expeditions. It aims to help the interpretation of measurements from detailed process studies of aerosols, clouds, and energy fluxes observed during AO2018, and gives insight into the very distinct cloud characteristics in the central Arctic during summer 2018.

## 2 The Expedition

AO2018 took place on the Swedish icebreaker Oden, between 1 August and 21 September 2018, departing from and returning to Longyearbyen. The expedition track and principal measurement stations are shown in Fig. 1. Oden entered the sea ice on 2 August, conducting a 24-hour measurement station within the Marginal Ice Zone (MIZ) (82.1547°N, 9.9695°E, from 23:00 UTC) before making its way toward the North Pole. A measurement station was undertaken at the closest point to the pole achievable (89.8932°N, 38.0423°E). At about 20:00 UTC on 13 August the Oden moored to a large, stable ice floe on which to undertake measurements, and drifted with it until 21:00 UTC on 14 September. A final 24-hour measurement station was undertaken on 20 September, within the MIZ (82.2833°N, 19.8333°E) before leaving the ice.

The meteorological component of AO2018 combined two projects: Microbiology-Ocean-Cloud Coupling in the High Arctic (MOCCHA) and Arctic Climate Across Scales (ACAS). The projects shared many measurements and operated jointly during the expedition.

## 3 Measurement systems

An overview of the measurement systems is given in Table 1. A suite of remote sensing instruments operated almost continuously throughout the expedition, providing a mobile Cloudnet (Illingworth et al. 2007) site. A Metek MIRA-35 scanning Doppler cloud radar was installed on the roof of a container on Oden's foredeck, a HALO Photonics Stream Line scanning micro-pulsed Doppler lidar (Pearson et al., 2009) was installed within a motion-stabilised table (Achtert et al. 2015) on top of a container above the foredeck laboratory. A Radiometer Physics HATPRO scanning microwave radiometer was



installed alongside the lidar. Radiosondes (Vaisala RS92) were launched from the ship's helipad every 6 hours (00:00, 06:00,

12:00, 18:00 UTC); data from these were shared globally in near-real time over the Global Telecommunication System (GTS). The measurements from these instruments allow a detailed characterisation of clouds using the Cloudnet algorithm. Cloudnet averages the data to a common grid at the cloud radar resolution and provides an objective hydrometeor target classification. Further products are derived on the basis of the hydrometeor target classification and the available measurements, including cloud occurrence, top and base height, cloud thickness, cloud phase, liquid water content, ice water content, and the effective

radius of cloud droplets and ice crystals. Details of the preliminary data processing steps required prior to running the Cloudnet retrieval and further information on the product retrievals are documented in Achtert et al. (2020).

Additional remote sensing measurements were made by a Campbell CS135 laser ceilometer, and a METEK MRR2 Micro Rain Radar, both installed above the foredeck laboratory. A Particle Metrics Forward Scattering Spectrometer Probe (FSSP-100) was installed above the container laboratories on deck 4 to measure the drop size distributions of fog. It was mounted on

a motorised rotator with a control system that monitored the local wind direction and kept the FSSP oriented into wind.

On the 7th deck, approximately 25 m above the surface a second ceilometer (Vaisala CL31) was installed, along with a weather station measuring pressure (Vaisala PTU300), temperature, and relative humidity (RH) (aspirated Rotronic MP101); wind speed and direction (heated Gill WindSonic M); and broadband downwelling solar and infrared radiation (Eppley PSP and PIR mounted on gimbals). A Heitronics KT15-II infrared temperature sensor measured the surface temperature. A present

weather sensor (Vaisala PWD22) measured visibility, precipitation type, precipitation intensity, and amount.

A turbulent flux system was installed on the foremast immediately above the bow at a height of 20 m above the surface. This consisted of a sonic anemometer (heated Metek uSonic-3), with an XSens MTi-G-700 motion pack to measure platform motion, a LI-COR LI-7500 infrared gas analyser to measure water vapour, and an aspirated Rotronic MP101 to provide a reference temperature and RH at the top of the foremast. Wind measurements are corrected for platform motion and for flow

distortion around the ship (Prytherch et al. 2015, 2017). Flux estimates were calculated via eddy covariance over 30-minute averaging intervals, and standard statistical quality control tests for skewness, kurtosis (Vickers and Mahrt, 1997) and stationarity (Foken and Wichura, 1996) were applied to flag unreliable estimates. Periods with flow from aft (wind directions more than 120° from the bow) are heavily contaminated by turbulence generated by the ship's superstructure, and were excluded. Such periods are very few, however, because the ship was re-oriented into the wind on a regular basis to maintain

clean sampling for the extensive aerosol measurements being made by other groups on board.

During the 4-week drift, additional measurement systems were installed on the ice floe. A 15 m mast was erected about 300 m from the ship with a heated sonic anemometer (Metek USA-100) at the top of the mast (15.55 m) and a Vaisala HMP-110 probe in an aspirated radiation shield just below the top of the mast, to measure temperature and RH. 4 more aspirated shields with T-type thermocouples, were mounted at approximately logarithmically spaced heights (0.80, 1.55, 3.05, 8.80 m) to

measure the near-surface temperature profile. A final thermocouple was buried at the ice/snow interface. NRG Type-40 cup anemometers were mounted at 5 levels (0.65, 1.45, 2.86, 6.65, 13.25 m) to provide a near-surface wind-speed profile. A second 2-m tall mast was located nearby with a Gill R3A sonic anemometer and a LI-COR LI-7500 gas analyser to make direct water



vapour flux measurements. About 50 m from the main mast, pairs of solar and infrared radiometers (Kipp & Zonen CMP22 pyranometer and CGR4 pyrgeometer) were installed to measure up- and down-welling radiative fluxes over an undisturbed

snow surface. Another Heitronics KT15-II measured the surface temperature immediately below the radiometers.

A second site was located at the edge of an open lead, approximately 1.5 km from the ship. A 2 m mast was instrumented with a Metek uSonic-3 sonic anemometer and two LI-COR infrared gas analysers: an open-path LI-7500 was used for water vapour measurements from which the latent heat flux was calculated, and a closed-path LI-7200 was used to make CO2 flux estimates. An aspirated Vaisala HMP-110 measured air temperature and RH, and a Heitronics KT15-II infrared temperature sensor

measured the skin temperature of the open lead surface.

## 4 General atmospheric conditions

Synoptic-scale atmospheric conditions exhibit large annual and interannual variability. To put the relatively short observation period from AO2018 in a larger context, prevailing conditions for 2018 are compared to climatology using NCEP Reanalysis data. Figure 2 shows mean sea-level pressure (MSLP) and its anomaly from the 1981-2010 climatology for the time of the

measurement campaign. There are two separate high-pressure areas, one over Greenland and one stretching from the Beaufort Sea over the East Siberian Sea to the Laptev Sea. Low pressure is centred over the Canadian Archipelago to the west and over the Barents Sea to the east, with the area around the North Pole, where the expedition took place, in-between these two low-pressure centres. The pressure pattern is anomalous compared to the 1981 to 2010 climatology, with a negative anomaly of more than 5 hPa over the Canadian Archipelago, 4 hPa over the Barents Sea, and around 1 hPa over the measurement location.

The positive anomaly over the Beaufort and East Siberian Sea was weaker than the negative anomalies, only 1-2 hPa.

The synoptic-scale weather development resulting from this large-scale setting is illustrated in Fig. 3. ECMWF surface-pressure, precipitation and 10-m wind charts are shown at weekly intervals through the ice drift, including the tracks for the five most significant low-pressure systems. The cyclonic activity seen here started in the middle of August and lasted until the end of the campaign in September. Earlier, the synoptic activity was weaker, with some weaker low-pressure systems

influencing the AO2018 track but also with some high-pressure influence (Fig. 4e). The first strong low-pressure system developed over the Barents Sea on 22 August (Fig. 3a), moving anticlockwise around the pole, bringing precipitation and enhanced wind speeds towards the location of the AO2018 ice drift (Fig. 4d, f). Two more low-pressure systems developed on 27 and on 31 August (Fig. 3b). One developed over the Kara Sea also moving anticlockwise around the pole and dissipating in the Canada Basin, whereas the other developed between Greenland and Svalbard, first moving eastwards and then turning

around towards the Kara Sea. These systems also affected the AO2018 ice drift, bringing precipitation and strong winds (Fig. 4d, f). The fourth low-pressure system moved from the Laptev Sea on 7 September (Fig. 3c) towards the Beaufort Sea and the last system developed over the East Siberian Sea on 12 September (Fig. 3d), moved towards Svalbard and then towards the Canadian Basin.



### 4.1 Near-surface conditions

Measurements of near-surface conditions undertaken on board the ship are shown in Fig. 4. The net surface energy was calculated from radiation measurements on board the ship and on the ice floe together with turbulent flux measurements from the foremast. Upward radiative fluxes were only directly measured on the ice. For the ship based radiation measurements, the upwelling longwave radiation was calculated using blackbody radiation from the KT15 surface temperature measurements, assuming an emissivity of unity. The shortwave upwelling radiation was calculated using 3 hourly albedo estimates made from

surface images of the surrounding of the ship. All fluxes are defined positive if they are directed towards the surface. Hence, a positive net surface energy flux represents energy input into the surface.

Wind speeds measured at the foremast varied between 0 and 13 m s$^{-1}$ (Fig. 4d). The strong variability was caused by the passage of the aforementioned low-pressure systems. The time series of near-surface temperature shows the transition between melt and freeze season (Fig. 4a). From the beginning of the campaign until 28 August, surface and air temperatures were

mostly between 0°C and –2°C with brief cooler periods of 1 to 2 days, usually with occurrences of clear skies. This is representative of the sea ice melt season when net positive surface energy acts to melt snow and ice but cannot warm the surface above the freezing point whilst the melting ice and snow remains. From 23 August onward temperatures gradually cooled and with another sudden drop on 28 August stayed below 0°C, mostly below –2°C, with a minimum surface temperature of –18°C, also in a cloud free period. An often-used definition for the onset of the freeze up is the time when the running-mean

near-surface air temperature falls below a certain threshold (Colony et al. 1992, Rigor et al., 2000; Tjernström et al., 2012). Here we follow Tjernström et al. (2012) using a threshold of –2°C, which determines the start of the freeze up as 28 August (Fig. 4a). The five days before 28 August show a slow transition between melt and freeze conditions, with the surface undergoing multiple freezing and melting cycles. The freeze onset can also be defined using the surface energy budget. A surplus in surface energy melts the ice and negative values indicate freezing. As can be seen in Fig. 4b surface energy drops

below 0 W m$^{-2}$ on around 23 August, but recovers briefly to above zero on 27 August. From 28 August onwards it stays below 0 W m$^{-2}$ coinciding with the freeze onset defined previously. Therefore, we will refer to the measurement period before the 28 August as the melt period and after as the freeze up period.

The near-surface atmosphere was very moist throughout the campaign. RH with respect to water (RHw) was mostly between 90 and 100% (Fig. 4c). Only at the beginning of the expedition until 17 August RHw was more variable ranging between 80

and 100%. RH with respect to ice (RHi) was in the same range until the freeze up. After 28 August RHi continued to be close to 100% with a slight supersaturation, consistent with Andreas et al. (2002), while RHw declined from 4 September onwards (Fig. 4c). This change in near-surface relative humidity is also reflected in the visibility (Fig. 4f). The visibility is more often limited, often < 1 km (fog), during the melt. When the surface becomes saturated with respect to ice and more precipitation falls as snow, the visibility is higher, since fog droplets tend to evaporate or deposit on the surface.

The probability distributions of ice surface temperatures peaked in the range between –1.8 and 0.0°C (Fig. 5a), the freezing points of sea and fresh water respectively, representing the conditions during the melt period. A secondary peak at –3°C and

the long tails towards colder temperatures represent the freeze up. For near-surface air temperature the distributions also peak between -1.8 and 0.0°C (Fig. 5b), which reflects the strong surface control on near surface air temperature during the melt seasons and the colder temperatures mostly reflect the freeze period. The slight differences between the ship and the ice station

measurements result from data gaps in the ice station time series, removing the additional ship data and creating a like-for-like comparison removes almost all the difference. As the near-surface atmosphere was very moist, the distribution for the RHw measurements peaks between 95% and 98% for the ship based measurements (Fig. 5c). The measurements from the ice station have peaks at 94% and 99%. The probability distributions for the wind speed peak at 4.5 m s$^{-1}$ for the ship measurements and 6.5 m s$^{-1}$ for the ice station (Fig. 5d). The differences again come from data gaps in the ice station time series. All distributions

have a tail of higher speeds, reaching 16.0 m s$^{-1}$. For the ice drift period both measurement sites show a higher probability of wind speeds above 9.0 m s$^{-1}$ reflecting the stronger synoptic activity during the ice drift period.

### 4.2 Surface fluxes

The turbulent fluxes were small as expected. The sensible heat flux calculated from the ship measurements peaked between -10 and 0 W m$^{-2}$ with tails for both the whole campaign and for the ice drift period only towards -20 and 10 W m$^{-2}$ (Fig. 5e).

The distribution for the ice station peaked at -1 W m$^{-2}$ with tails towards -20 and 10 W m$^{-2}$. The latent heat flux peaked at -5 W m$^{-2}$ for the ice and ship station, with a wider tail towards negative values around -20 W m$^{-2}$ (Fig. 5f). The distributions of net shortwave radiation peak around 9 W m$^{-2}$ with a long tail towards 70 W m$^{-2}$ for both the whole measurement period and the ice drift (Fig. 5g). Net longwave radiation peaks at around -5 W m$^{-2}$ with a similarly long tail towards -70 W m$^{-2}$ (Fig. 5h). The total surface energy budget distributions are very similar for the ice and ship measurements, peaking at -10 W m$^{-2}$ and -

15 W m$^{-2}$, respectively (Fig. 5i). All distributions have a tail towards -60 W m$^{-2}$ and shorter tail towards positive values.

### 4.3 Vertical structure

Time-height cross sections of equivalent potential temperature, wind speed and RHw measured by radiosondes give an overview of the vertical structure of the atmosphere during the expedition (Fig. 6a-c). Additionally, cloud target classification from the Cloudnet algorithm gives an overview of cloud cover and cloud phase for the same time period (Fig. 6d). The thermal

structure shows a gradual cooling and reduction of stability over time. Several frontal systems affected the measurement area during the campaign. These systems were associated with deep frontal clouds and strong winds throughout the whole troposphere. Between those high-wind periods, wind speeds were not only low within the surface mixed layer (SML) but also aloft. RHw was high within the SML. Aloft, RHw was very variable. Within frontal systems, RHw was high throughout the whole vertical column whereas it dropped to below 30% above the SML on several other occasions. RHi (not shown) shows

a gradual descent in altitude of the saturation level over time, from around 2 to 3 km in mid-August to close to the surface in early September.

The probability distributions of equivalent potential temperature and RHw as a function of altitude show that there are two predominant structures occurring in the vertical thermodynamic profiles (Fig. 7). One with a well-mixed deep layer up to ~1.5





km and the other with a shallower well-mixed layer reaching 400 to 500 m. Both are capped by a temperature inversion. The
well-mixed near-surface layers appeared preferentially at an equivalent potential temperature between 8.0 and 10.0°C or 4.5°C
and were very moist with relative humidity between 90% and 100%. The RHw distribution shows a very moist layer with
humidity above 90% up to about 800-1000 m.

The characteristics of the main temperature capping inversion are shown in Fig. 8. To identify the main capping inversion
from the radiosonde profiles an objective algorithm is applied to the temperature and equivalent potential temperature profiles
in a decision-tree-like process mostly following Tjernström and Graversen (2009). To summarise, all layers with a positive
temperature gradient deeper than 20 m within the lowest 3 km are identified and layers separated by less than 100 m are
merged. The layer with the strongest gradient is considered the main inversion. If no temperature inversion could be identified
the strongest stable layer within the lowest 3 km that was at least 20 m deep and 0.1 K strong was identified using the equivalent
potential temperature profiles and used as a proxy for the main inversion. The main inversion base is used as a proxy for the
boundary layer (BL) height. If there are weaker inversions below the main inversion, the lowest inversion base is considered
to be the height of SML and the rest of the BL is considered to be decoupled from surface induced turbulence.  In a similar
decision-tree like processes as for the main inversion, the strongest stable layer below the main inversion was considered as
the SML when no weaker temperature inversion could be found. In addition to the radiosonde data, surface temperature
measurements from the KT15 on board the ship were used to identify surface inversions. If the temperature was monotonically
increasing from the surface to the lowest measurement heights of the radiosonde (30 m) it was classified as a surface inversion.
In these stable conditions surface processes are also decoupled from the rest of the BL.

The analysis for all available radiosondes revealed that the BL was coupled for 41.0% of the time and decoupled for 59.0%.
From those 59.0% decoupled cases 13.5% were decoupled by a surface inversion and the other 45.5% by a weaker inversion
below the main capping inversion. The probability distributions of the capping inversion and SML characteristics are shown
in Fig. 8. The main capping inversion base height shows a bimodality with a maximum below 400 m and another one around
1500 m. The inversions were mostly 50 to 300 m thick and the inversion strength shows a broad distribution of 1.0 to 8.0 K,
with a maximum at around 1.5 K.  The SML was mostly between 50 and 400 m deep.

### 4.4 Cloud characteristics

Cloud characteristics and cloud phase are determined on a profile-by-profile basis using the Cloudnet target classification
(Illingworth et al., 2007) with a temporal resolution of 30 s following Achtert et al. (2020). For the entire measurement period
94102 profiles are available. From these profiles, only 4% detected no clouds, 41% had a single cloud layer and 54% multiple
cloud layers. Profiles of cloud fraction per volume (Brooks et al. 2005) have been obtained using time-height sections of 30
min and 90 m height. As shown by Achtert et al. (2020), the target classification reveals an unrealistically high occurrence of
the targets Aerosol, Aerosol & insects, and Insects during periods that were actually dominated by fog. Hence, we follow their
approach and re-classify the targets for these categories as fog during periods with visibility < 1 km. Note, however, that the
radar's lowest range gate is at 156 m and consequently many of the shallower fog episodes were missed by the radar. For





AO2018 visibility data shows that 49% of the fog occasions were too shallow to be detected by Cloudnet, so low level liquid clouds are likely underestimated.

Fog depths could still be calculated using radar range-height indicator (RHI) scans. Radar reflectivity was averaged between
150 and 1000 m away from the radar to obtain mean vertical profiles of radar reflectivity. The fog layer top height was defined as the strongest negative vertical reflectivity gradient in the lowest 500 m. If there were several cloud layers in the first 500 m the strongest gradient in the lowest layer was used for the fog depth. As radar reflectivity is proportional to the drop diameter to the power of 6, light precipitation or drizzle can be expected to influence the results for higher reflectivity. Hence, the micro rain radar data was used to reject all detected fog heights during precipitation events. Fog, defined as visibility < 1km, was
detected approximately 21% of the time during AO2018 (Fig. 9a). The probability distribution of the fog depths is shown in Fig. 9b. The most common fog depth was between 120 and 150 m, indeed below the lowest range gate of the radar, with a median of 205 m and a tail extending just above 500 m.

Calculated cloud occurrence probability distributions as a function of height are shown for the entire campaign, the melt and the freeze period in Fig. 10. Cloud fraction was largest below 1 km for the entire campaign and separately for both melt and
freeze up periods. This is reflected by a maximum cloud fraction of approximately 65% for the entire campaign, 49% for the melt period and 69% during freeze up, occurring below 500 m. A secondary maximum appears between 2.5 and 3.0 km for the total and for the melt period distribution. A third maximum appears for the melt period at around 4.5 km. During the freeze up period the secondary maximum was higher, at 3.0 to 4.0 km. These secondary maxima reflect the frequent occurrence of multiple cloud layers during AO2018. Mixed phase clouds were the most abundant cloud type, occurring below 4 km during
the melt period and below 3 km during the freeze up period. Above these levels ice clouds dominated. However, some mixed phase clouds were observed up to a height of 8 to 9 km. Liquid-only clouds were rarely observed in either season, even for the low altitudes. Note, however, that liquid water clouds occurring below the lowest radar range gate (fog) are not included here. Statistics of cloud top, base and thickness are shown in Fig. 11 for the lowest two cloud layers separated by cloud phase. A limitation of the Cloudnet approach is that there is no distinction between falling ice particles and cloud ice. Hence, ice
precipitation extends the apparent cloud boundaries. Furthermore, results flagged as one cloud might actually contain two cloud layers with ice precipitating from the upper clouds into the lower. This might be the case for the thicker ice and mixed phase clouds in particular. Based on an analysis of cloud radar Doppler spectrographs from ASCOS, Sotiropolou et al. (2014) suggested that a mixed phase cloud depth of over 700 m might be considered two cloud layers, consistent with this hypothesis. These limitations should be kept in mind for comparisons with other observational results not obtained with the Cloudnet
algorithm. For a comparison with model results, this might not cause problems, as some models treat falling ice particles the same way as the Cloudnet algorithm or ice precipitation can be included for statistics.

For the first cloud layer, mixed phase clouds were detected in 47% and 52% of the profiles during melt and freeze, respectively. During melt another 19% of the lowest clouds were identified as liquid clouds and 34% as ice clouds. For the freeze period only 10% were liquid clouds and 38% were ice clouds. The results for the lowest cloud layer show that all of the clouds have
very low cloud bases, with median cloud bases at 180 to 200 m; including the lowest clouds below the radar's lowest range





gate this is probably even lower. Liquid clouds are by far the thinnest clouds with a median thickness of 72 m, while ice clouds have a median thickness of around 400 to 450 m for both seasons. However, their vertical extent is quite variable as indicated by the much higher mean cloud thickness and the extent of the 75[th] and 95[th] percentile. First-layer mixed-phase clouds are considerably thicker than ice clouds, with a median thickness of 1500 m and 2280 m for the melt and freeze periods,

respectively.

In the case of multiple cloud layers, 71% of the second layer clouds were ice clouds during melt and 80% during the freeze period. Second layer ice clouds have a much higher cloud base than second layer mixed and liquid clouds, with a median cloud base height of around 2800 m during melt and 3500 m during freeze compared with 800 m for mixed phase clouds in both seasons and 1500 m and 800 m for liquid clouds during melt and freeze, respectively. The thickness of liquid second-layer

clouds is, as for the first layer, very thin, with a median value of 72 m during melt and 120 m during the freeze period. Second layer ice clouds have a similar thickness as first layer ice clouds, but second layer mixed phase clouds are much thinner than mixed phase first layer clouds with median values of 672 and 863 m for melt and freeze, respectively.

## 5 Temporal evolution

For a more detailed analysis of the meteorological conditions, the thermodynamic structure was used to divide the campaign

into 8 distinct periods (Fig. 4 and 6). Periods were defined by similarity of equivalent potential temperature and RHw profiles. Period 1 covers the time in the MIZ until 4 August 06:00 UTC. Period 2 encompasses the journey into the ice towards the North Pole until 12 August 00:00 UTC. Since cloud radar measurements were not possible during heavy ice breaking because of excessive vibration, cloud characteristics and fog heights are not available during period 2. Period 3 (12 to 17 August) includes the 'North Pole' station and the beginning of the ice drift. Period 4 (18 to 27 August) covers the end of the melt and

the transition period into the freeze up. The freeze up is covered by period 5 (28 August to 3 September), 6 (4 to 7 September) and 7 (8 to 12 September 12:00 UTC). Finally, period 8 (12 September 12:00 UTC to 21 September 06:00 UTC) covers the end of the ice drift period and the transit out to the ice edge.

### 5.1 Near-surface development

The time in the MIZ (P1) shows surface temperatures still above 0°C and the air was saturated (Fig. 4 and 6). Periods P2 to P4

were typical for the melt season within the central pack ice. Near surface air and ice surface temperatures were around 0°C, with short periods of lower temperatures, in particular during P3 (Fig. 4a). The lower temperatures were caused by a high pressure system, resulting in cloud free conditions, which reduced the downwelling longwave radiation and resulted in a temporarily negative net surface energy budget and cooling of the surface and near surface temperature. RH was quite variable during these periods, in particular during P3, corresponding to the changes in cloud conditions (Fig. 6). Towards the end of P4

temperature started to drop below 0°C indicating the transition towards freezing conditions. A further drop in temperature on 28 August marks the beginning of the freeze up and the start of P5. During P5 and P6 temperatures were mostly below -2°C



with short much colder periods corresponding, again, to cloud free conditions (Fig. 6). Most of the time P5 and P6 were under the influence of three strong low-pressure systems passing over the regions (Fig. 3b, c), resulting in strong winds and a considerable amount of precipitation (Fig. 4d, f). P7 was the coldest period during the ice drift and was dominated by high

pressure (Fig. 4d), which caused lower wind speeds and mostly cloud free conditions without any precipitation (Fig. 6). The last period (P8) had again quite variable conditions with temperatures ranging between -2 and -14°C as it was influenced by both low and high-pressure systems resulting in periods of stormy conditions with precipitation and high winds, and calm cloud free conditions (Fig. 6).

**5.2 Thermodynamic development**

The vertical structure over the MIZ (P1) shows distinct differences from the other periods within the central pack ice. While in the MIZ, the air was coming from the south-east, advecting warm air over the melting sea ice towards the location of the ship, resulting in a stably stratified air mass (Fig. 12a). The vertical profile of the equivalent potential temperature shows a strong stratification in the lowest 150 m, followed by a layer of weaker stratification up to around 650 m. The inversion statistics show that this period was dominated by strong and deep surface inversions (Fig. 13). Within the central pack ice, the

thermodynamic structure of the atmosphere gradually changed from P2 to P3, with a reduction in stability (Fig. 12a) and a slight increase of the main inversion base height, but the inversions remained quite strong and deep (Fig. 13). P4 and P5 show quite distinct inversion characteristics. The median inversion base height is nearly 1000 m higher than in P3 (Fig. 13) but inversions were thinner and weaker. This is most likely caused by the strong synoptic activity during these periods with several frontal systems dominating the thermodynamic structure of the atmosphere. P5 was also the period with the highest wind

speeds (Fig. 12c). P6 was partly influenced by a low pressure system, also causing higher wind speeds (Fig. 12c) and most likely causing the wide spread of inversion base heights (Fig. 13). P7 was influenced by high pressure, resulting in more stable and very cold conditions (Fig. 12a). The main capping inversions were rather strong and low with a median base height of about 200 m. The final period has a low median main inversion base height of about 400 m, but shows high variability as indicated by the 25[th] and 75[th] percentile (Fig. 13). It is also warmer than the previous period, and has higher median wind

speeds.

One important characteristic of the Arctic BL, particularly in the summer, is a frequent decoupling of the SML and the cloud mixed layer (CML) (Shupe et al. 2013, Brooks et al. 2017), a feature that models generally fail to represent (Birch et al. 2012; Sotiropoulou et al. 2016). This is in particular relevant when investigating local aerosol production as a source of CCN or INP and their impact on cloud. These particles can only affect cloud properties if they are mixed up to the clouds. Surface processes

can be decoupled from the clouds via a secondary weak inversion below the main inversion or by a surface inversion, i.e. stable conditions. An overview of the relative amount of coupling and decoupling and the respective process of decoupling is listed in Table 2. P1 was mostly decoupled by surface inversions. P2 and P3 were decoupled for around 45% of the time, where P2 still experienced a lot of surface inversions, whereas during P3 no surface inversions were observed. The median SML height was 155 and 170 m for P2 and P3 respectively, but the spread of SML heights was much wider for P2, reaching down





to 0 m representing the frequent occurrence of surface inversions (Fig. 13a). P4 and P5 were decoupled for 82% and 69% of
the time, respectively, with a much deeper SML than during the previous periods (Fig. 13a). Conditions during P6 were quite
different and the BL was decoupled only 40% of the time with rather deep SMLs. The quite cold period P7 was decoupled
53% of the time with particularly shallow SMLs. The median SML height was only 75 m. The last period was decoupled 48%
of the time with one third of the decoupling caused by a surface inversion.

### 5.3 Cloud characteristics

Frequency of occurrence of single-layer and multilayer clouds for each period are shown in Fig. 14. Cloud occurrence
probability distributions are shown in Fig 15, and statistics of cloud top, base, and thickness for the first two layers in Fig 16.
During P1 multiple cloud layers were present for 80% of the time and single-layer cloud occurrence was dominated by mixed
phase clouds. All clouds below 2.5 km were liquid clouds, while above this level mixed phase and ice clouds reached up to

9.0 km (Fig. 14). The cloud layer statistics for P1 (Fig. 16) show these ice and mixed phase clouds to be deep if they are the
first cloud layer; they most likely consist of several cloud layers with precipitating ice in between. These were predominantly
precipitating frontal clouds (Fig. 6d).
During P3 nearly 60% of the Cloudnet profiles had a single cloud layer and about 14% showed no cloud layer (Fig. 14). There
is also a very low cloud fraction per volume for all heights, with a total maximum of 30% in the lowest 500 m and below 20%

higher up (Fig. 15). First layer clouds were mostly either shallow liquid and ice clouds or deeper mixed phase clouds (Fig. 16).
For times with multiple cloud layers, the statistics for the second cloud layer show very thin liquid clouds at about 2.4 km
height, thin ice clouds with a median cloud base height of 3.2 km or low level deep mixed phase clouds. P3 is the only period
showing a second layer of liquid clouds with such predominantly high cloud bases.
P4 was influenced by a low-pressure system moving anticlockwise around the measurement location (Fig. 3a). This resulted

in multiple cloud layers for about 68% of the time and much higher cloud occurrence than in P3 with a maximum cloud fraction
of 70% below 1000 m and a secondary maximum of 30% at around 4500 m (Fig. 15). The cloud statistics for P4 show that
liquid clouds are very thin and occur predominantly below 1.1 km with a median cloud thickness of 72 and 120 m and cloud
tops of 300 and 1070 m for first and second layer clouds, respectively. First layer mixed phase clouds have a median cloud
base at 215 m and a median cloud top at 1835 m and second layer clouds have a median cloud base at 730 m and cloud top at

1580 m. Mixed phase and liquid clouds predominantly occur below 2 km and clouds at higher altitudes are predominantly ice
clouds. Ice cloud statistics show a large difference between first and second layer clouds. First layer clouds have a median
cloud base and top of 180 m and 1190 m, respectively, whereas second layer ice clouds have a large variability of cloud bases
and tops with median values of 2700 m and 3730 m.
P5 and P6 show similar cloud characteristics. Cloud fraction per volume was over 70% below 1 km for P5 and over 80% for

P6, which is the highest occurrence frequency of all 8 periods (Fig. 15). Also cloud fraction above 1 km was high, more than
60% up to 5 km. Mixed phase clouds dominated up to 4.5 km in P5 and 3.5 km during P6. Statistics of cloud base, top and
thickness are comparable for P5 and P6. First layer mixed phase clouds had a median thickness of 4100 m. They were thicker





than liquid and ice clouds and also thicker than during the other periods, except P1. First layer ice clouds were also thicker than during the other periods, with median values of 1700 m. Second layer ice clouds show higher median cloud bases compared to the other periods, at 4380 m and 3995 m for P5 and P6, respectively.

During P7 multiple cloud layers were present for nearly 70% of the time. Cloud fraction peaks at around 3.5 km with a secondary maximum below 1.0 km (Fig. 14). Overall, first layer clouds were thinner than in P4 to P6 but second layer clouds, in particular ice clouds were thicker than in previous periods with a median value of 815 m.

For P8 statistics show that no clouds were detected for about 10% of the time and about 47% of the time single layer clouds were present. The highest cloud fraction was observed in the lower levels, at 300 m with a secondary maximum of about 30% at 4 km. Clouds, in particular first layer mixed phase clouds, were rather thin compared to the other periods with a median cloud thickness of 840 m, and second layer ice clouds had a lower cloud base than during the rest of the freeze periods (P5 to P7).

Figure 17 shows the relative amount of fog occurrence during each period and the respective fog depths. Period 1 was the foggiest with visibility below 1 km for 70% of the time, followed by P2 with fog present around 50% of the time. Period P3 and P4 had much fewer fog episodes, with 27% and 22%, respectively. P5 was mostly fog free, and P6-P8 had fog around 12% to 15% of the time. Fog depths are shown in Fig. 17b. These are quite similar throughout the measurement campaign, showing slightly higher median depths for P1-P4, than for P5-P7. P8 shows unusually deep fog layers with a calculated fog depth of over 400 m. However, radar data was only available for the first three days of P8 and only 10% of the fog occurred within these 3 days. Hence, the calculated fog depth of over 400 m may not be representative of the total fog conditions during P8.

## 6 Comparison with previous campaigns

Here we compare the AO2018 observations with those from previous campaigns, providing insight into common features and significant differences. We compare the 2018 observations with the expeditions AOE96, AOE2001, ASCOS, and AO2016, as these campaigns all operated in the central Arctic Ocean during the melt/freeze transition. Additional comparison is also made with results from SHEBA in the Chukchi Sea during August and September 1998 and ACSE along the Siberian Shelf in 2014 where possible.

From the large-scale perspective the conditions during AO2018 are most similar to those of ASCOS, with low pressure over the Canadian Archipelago and over the Barents Sea (Tjernström et al., 2012). However, the strong high-pressure centre over the Canada Basin was absent in 2018. Instead of being in a clear anticyclonic circulation, as during ASCOS, or a clear cyclonic flow from one low pressure centre, as during AOE2001, AOE96, and SHEBA (Tjernström et al., 2012), AO2018 sat between two low-pressure areas. As a result, several low-pressure systems propagated westward around the pole, influencing the AO2018 measurement campaign. A clear difference to ASCOS is the timing of the low-pressure systems. During ASCOS most of the storms happened in early and mid-August, at the beginning of the campaign and the installation of the ice camp,



while the later ice drift period was in rather calm conditions. During AO2018 the low-pressure systems passed over the measurement site throughout the campaign with most of the strong low-pressure systems influencing the ice drift measurements from mid-August into September.

This strong synoptic activity during AO2018 resulted in a vertical structure of the atmosphere that differed to the earlier campaigns. The vertical probability distribution of the equivalent potential temperature (Fig 6a) shows two predominant BL

depths. One near-neutrally stratified layer up to 400 to 500m, and another near-neutrally stratified layer up to ~1.5 km. In contrast, the results from the previous campaigns were very consistent with only one dominating mixed-layer height of about 300-400m (Tjernström et al., 2012). The AO2018 moisture profile is, however, consistent with earlier campaigns showing a layer with very high RH up to about 800 to 1000 m (e.g. Tjernström et al., 2005, Sedlar et al. 2011, Devasthale et al. 2011).

To better compare the inversion characteristics, statistics were calculated using radiosonde profiles for all available campaigns

(Fig. 18). The median heights of the main capping inversion bases for all campaigns are in the range of 310 to 570 m, with ASCOS having the highest median inversion base heights. AO2018 has a much wider distribution than the other campaigns, with the 25th and 75th percentile at 135 m and 1500 m, respectively. The other campaigns third quartile ranged from 860 m to 1200 m. AO2018 inversions were shallower than during AOE2001, ASCOS, and ACSE, but comparable to the other two and the inversion strengths were comparable to AO2016, AO2001, and SHEBA.

Another typical feature of Arctic BLs is the decoupling of the SML from the CML. This means that an upward transfer of heat, moisture and aerosols from the surface up to the clouds aloft is often inhibited. During AO2018 decoupling with a distinct SML was observed 45% and surface inversions 14% of the time. The rest of the time the BL was coupled. For comparison, Brooks et al. (2017) found the ASCOS boundary layer to be decoupled 48% of the time during the ice drift, and 76% of the time during a period with a single deck of stratocumulus. Sotiropoulou et al., (2014) found similar results for a longer cloud-

covered time period, with a decoupling frequency of 72%. If all available radiosonde profiles for the ASCOS campaign are considered (i.e. including the transit into and out of the ice) the decoupling frequency is 57.2%, similar to the frequency of decoupling observed during ACSE (56.9%), AOE2001 (55.1%) and SHEBA (57.3%). AO2016 had less frequently decoupled boundary layer conditions with only 45.2%.

The near-surface conditions during AO2018 were similar to those observed during the other campaigns as analysed in

Tjernström et al. (2012). Before the freeze up, temperatures were around 0°C with occasional brief cooler periods, mostly resulting from cloud-free conditions (Tjernström et al., 2005; Sedlar, 2011). RH, wind speeds, and visibility were also similar to the ranges observed in the other campaigns. The date of the start of freeze-up agrees well with other studies in the central Arctic, showing that the freeze up occurs in the second half of August or early September (e.g. Rigor et al., 2000, Overland et al. 2008, Tjernström et al. 2012, Sedlar et al., 2011). The surface fluxes are also generally similar to those observed during the

other campaigns, in particular ASCOS, but there are notable differences to the AOE96, AOE2001, and SHEBA incoming solar radiation distributions. AO2018 had a pronounced peak at 50 W m$^{-2}$ and only a few cases with higher solar radiation up to 250 W m$^{-2}$, whereas AOE96, AOE2001, and SHEBA peaked at higher values and had a wider distribution, this might be caused by the slightly different campaign durations and locations. AO2018 was longer than most of the summer campaigns and, hence,



had more cases with low incoming radiation. Furthermore, SHEBA was located further south and the smallest daily solar
zenith angles therefore were smaller, resulting in a wider distribution with higher values.

As well as the difference in the thermodynamic vertical structure, cloud characteristics for AO2018 differed from former campaigns. The vertical cloud fraction distribution for AO2018 (Fig. 9) showed a maximum below 1 km, similar to ACSE and ASCOS (compare Fig. 9 in Achtert et al., 2020). The peak for AO2018 was, however, not as pronounced as during ASCOS. AO2018 had a near absence of liquid clouds and a much smaller number of mixed phase clouds than during previous
campaigns. Another difference during the freeze period was the much higher cloud fraction between 1 and 4 km. This was above 50% during AO2018 but only ~30% during ASCOS and 10% during ACSE (Achtert et al., 2020). This can most likely be attributed to the multiple low-pressure systems passing the AO2018 track during the second half of the campaign, bringing deep-reaching frontal cloud systems. Compared to ACSE and ASCOS, the much higher fraction of ice clouds between 1 and 4 km is noticeable, which could result from secondary ice formation due to seeding of the lower clouds from falling ice
precipitation from higher clouds. The few liquid clouds were considerably thinner than during ACSE, with a median depth of 95 m during AO2018 compared to 220 m during ACSE (Achtert et al., 2020). This might be attributed to the location of the campaigns, with ACSE being farther south than AO2018, leading to overall warmer temperatures and more open water, and hence more favourable conditions for liquid clouds. Ice clouds were considerably thicker than during ACSE, with median values of 400 to 600 m compared to around 250 m during ACSE (Achtert et al. 2020).

Fog occurred 21% of the time during AO2018, less than during ASCOS (25%), but more often than during AOE2001 and AOE96 (10-15%, Tjernström et al., 2012). Fog depths were studied for the first time here and, hence, cannot be compared the other campaigns.

## 7. Summary and Conclusions

This paper provides an overview of the atmospheric measurements and conditions during AO2018, which took place on the
icebreaker Oden in the central Arctic Ocean from 1 August until 21 September 2018. The results are also compared with those of previous Arctic field campaigns from the summers of 1996, 1998, 2001, 2008, 2014, and 2016.

The large-scale atmospheric conditions had the campaign under the influence of two low-pressure areas. One was centred over the Canadian Archipelago to the west and the other over the Barents Sea to the east, with the AO2018 track located in the middle. This resulted in several synoptic storms passing over the AO2018 track, in particular from mid-August until the end
of the campaign in late September. AO2018, like previous campaigns, featured a moist, near-neutrally stratified BL, however there were two distinct regimes in vertical structure. One featured a well-mixed BL up to about 300 to 400 m while the other showed a well-mixed layer up to about 1500 m. This is also represented in the wide spread of the inversion base heights. 50% of the inversion bases were below 370 m, another 25% were considerably higher, up to 1500 m, and the rest even higher. The humidity profiles showed only one regime, similar to previous studies with a kilometre-deep moist boundary layer. The
boundary layer was decoupled 59% of the time.





In contrast to the vertical structure, near-surface conditions shared the same common features as previous campaigns. During the melt period the near-surface temperature was mostly between -2 and 0°C. After the start of the freeze up, around 28 August, temperatures decreased, reaching a minimum of about -15°C. The near-surface atmosphere was very moist with RH mostly above 90%. Near-surface winds were mostly between 2 and 7 m s$^{-1}$, but occasionally reaching up to 16 m s$^{-1}$ during the passage

of low pressure systems. Surface energy fluxes were similar to the range observed in previous campaigns. Net shortwave radiation peaked at 10 W m$^{-2}$ with a positive tail. The peak corresponds to the mostly cloudy conditions and the tail reflects the few cloud free conditions. The net longwave radiation PDFs peak at -5 W m$^{-2}$, having a long tail to smaller values. Turbulent fluxes were as expected very small, peaking at -10 to 0 W m$^{-2}$ for sensible heat and -5 W m$^{-2}$ for the latent heat flux.

The cloud occurrence was high throughout the campaign, dominated by low level clouds but with a substantial amount of mid-

level clouds. In particular during the freeze up the cloud fraction was above 50% between 1 and 4 km, much higher than for the earlier expeditions. The unexpected high occurrence of multiple cloud decks and the absence of prolonged periods with shallow, single layer stratocumulus clouds is most likely attributed to the strong cyclonic activity throughout the whole campaign. Several weaker low-pressure systems influenced the AO2018 measurements in the first half of August and multiple strong low-pressure systems occurred in the second half of August associated with frontal cloud systems until the end of the

measurement campaign in September. This also resulted in a strongly reduced occurrence of liquid clouds. Most of the clouds observed were either ice or mixed phase clouds. The lowest cloud layer was dominated by mixed phase clouds, while during times with multiple cloud layers, the second cloud layer was dominated by ice clouds. Cloud thickness depended strongly on the cloud type with a median cloud thickness of the lowest two layers of only 95 m for liquid clouds and 530 m for ice clouds. For mixed phase clouds, thickness also varied strongly between first and second layer clouds with 1700 m for first layer mixed

phase clouds and 740 m for second layer clouds. However, what cannot be ruled out is that the statistics for mixed phase and ice clouds might contain multiple cloud layers with falling ice in between the cloud layers, detected as one cloud; the Cloudnet algorithm cannot distinguish between falling ice and cloud ice particles. Falling ice from higher clouds into lower cloud layers might also be responsible for the high amount of ice clouds at relatively low heights, as these ice particles might trigger secondary ice formation in the lower clouds. Visibility measurements indicated frequent occurrence of fog, but with very

variable persistence; fogs became somewhat less frequent during the freeze up. Analysis of radar RHI scans revealed that fog layer depths were predominantly less than 200 m.

Overall the meteorological results from AO2018 summarised here provide a guide for further investigation. In particular, the strong cyclonic activity and the associated changes of the thermodynamic structure, the cloud types and the vertical cloud distribution from previous years raises the question of whether this was an exceptional year or if these changes are

representative of climatological change in Arctic summer atmospheric conditions. Reanalysis data already shows an increase of Arctic cyclone activity during the second half of the twentieth century (Zhang et al., 2004) and global and regional climate models suggest a further increase of cyclone activity during summer over the Central Arctic by the end of the 21$^{st}$ century (Orsolini and Sorteberg, 2009; Nishii et al., 2015; Akperov et al., 2019).



***Data availability.*** UK contributions, as well as selected other data, are available within the associated data collection in the Centre for Environmental Data Analysis (CEDA) archives. Other cruise data may be available in the Bolin Centre for Climate Research MOCCHA/AO2018 holdings (http://www.bolin.su.se).

***Author contributions.*** JV has analysed the data set and prepared the manuscript together with IMB, MT, JP, and PA. IMB,
PA, MT, and JP performed the measurements during AO2018. All authors contributed to the discussion of the results and revision of the manuscript.

***Competing interests.*** The authors declare no competing interests.

***Acknowledgements:*** This research is part of the Arctic Ocean 2018 expedition and was funded by the UK Natural Environment Research Council (NERC) (grant No. NE/R009686/1). The cloud radar, HALO Lidar, RPG HATPRO radiometer, Campbell ceilometer, radiosounding station, and micro rain radar were all provided by the Atmospheric Measurements and Observations Facility (AMOF) of the UK National Centre for Atmospheric Science (NCAS). The ACAS deployment, including the weather station, radiation, visibility and ceilometer instruments and flux observations aboard Oden were supported by Knut and Alice
Wallenberg Foundation. The soundings were supported by Environment and Climate Change Canada in collaboration with the Year of Polar Prediction, Polar Prediction Project. Additional flux observations on the ice were supported by the Bolin Centre for Climate Research. The Swedish Polar Research Secretariat (SPRS) provided access to the icebreaker (I/B) Oden and logistical support in collaboration with the U.S. National Science Foundation. We would like to thank Capt. Mattias Petersen and the crew of Oden for their invaluable support throughout the field campaign. Thanks also to the Chief Scientists Caroline
Leck and Patricia Matrai for the planning and coordination of AO2018. All data are available from the Bolin Centre Database (bolin.su.se/data/) and the NERC Centre for Environmental Data Analysis (CEDA)(ceda.ac.uk).



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



**Table 1: Overview of meteorological instruments. All heights for instrumentation on Oden are given relative to the waterline.**

| Instrument System | Location | Variables | Date of operation |
|---|---|---|---|
| Scanning Doppler cloud radar (Metek MIRA-35) | Container roof Oden's foredeck (12m) | reflectivity, doppler velocity, spectral width, linear depolarization ratio | 02/08-04/08 12/08-19/09 |
| Scanning micro-pulsed Doppler lidar (HALO Photonics Stream Line) | Container roof Oden's foredeck laboratory (12m) | doppler velocity, backscatter coefficient | 01/08-12/08 14/08-20/09 |
| Scanning microwave radiometer (Radiometer Physics HATPRO) | Container roof Oden's foredeck laboratory (12m) | temperature profile, liquid water path, integrated water vapour | 02/08-14/08 17/08-20/09 |
| Radiosondes (Vaisala RS92) | Oden's helipad (14.5 m) | temperature, relative humidity (RH), pressure, wind speed and direction as a function of altitude | 02/08-21/09 |
| Ceilometer (Campbell CS135) | Above foredeck laboratory (9.5 m) | cloud base | 01/08-20/09 |
| Micro Rain Radar (METEK MRR2) | Above foredeck laboratory (9.5 m) | reflectivity, rain rate, liquid water content, fall speed | 01/08-20/09 |
| Forward Scattering Spectrometer Probe (Particle Metrics, FSSP-100) | Above the container laboratories on deck 4 | drop size distributions of fog | 01/08- 06/09 |
| Ceilometer (Vaisala CL31) | 7th deck (25 m) | cloud base | 01/08-05/10 |
| Weather station (Vaisala PTU300, Rotronic MP101, heated Gill WindSonic M, Eppley PSP and PIR) | 7th deck (25 m) | pressure, temperature, RH, wind speed and direction, broadband downwelling solar and infrared radiation | 01/08-05/10 |
| Infrared temperature sensor (Heitronics KT15-II) | 7th deck (25 m) | surface temperature | 01/08-05/10 |
| Present weather sensor (Vaisala PWD22) | 7th deck (25 m) | visibility precipitation type and intensity | 01/08-05/10 (vis) 13/08-05/10 (prec) |
| Turbulent flux system (Gill R3A, heated Metek uSonic-3, XSens MTi-G-700 motion pack, LI-COR LI-7500, Rotronic MP101) | Oden's foremast (20 m) | 3 wind components, sonic temperature, platform motion, water vapour, temperature, RH | 01/08-20/09 |
| Turbulent flux system (Gill R3A, LI-COR LI-7500) | Ice floe (2 m mast) | 3 wind components, sonic temperature, water vapour | 18/08-14/09 |
| Met station (Metek USA-100, HMP110, T-type thermocouples, NRG Type-40 cup anemometers) | Ice floe (15 m mast) | 3 wind components, sonic temperature, temperature, RH, wind speed, surface temperature, | 18/08-14/09 |
| Radiation measurements (Kipp & Zonen CMP22 pyranometer and | Ice floe (1.5 m mast) | up- and down-welling radiative fluxes, surface temperature | 18/08-14/09 |





| | | | |
|---|---|---|---|
| CGR4 pyrgeometer, Heitronics KT15-II) | | | |
| Micrometeorology (Metek uSonic-3, LI-COR LI-7500, LI-COR LI-7200, Vaisala HMP-110, Heitronics KT15-II | open lead (2m mast) | 3 wind components, sonic temperature, water vapour, CO2, temperature, RH, surface temperature | 16/08-12/09 |

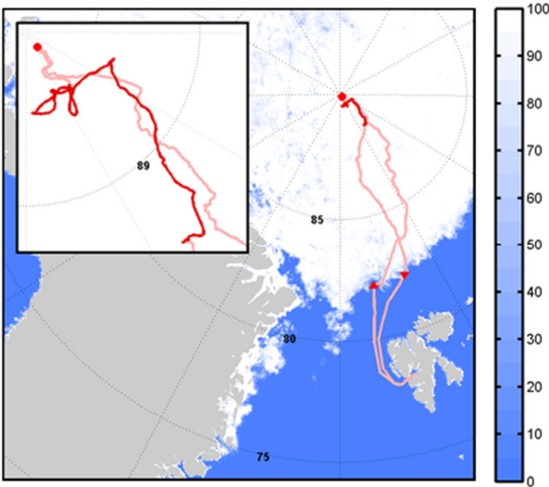

**Figure 1. Cruise track (pink) with the ice drift track (14 August – 14 September 2018, red and inset). The measurement stations within the marginal ice zone are marked by ▲ (inbound, 2 August 2018), and ▼ (outbound, 20 September 2018), and the 'north pole' station by • (12 August 2018). Colour gradient shows ice concentration (%) for September 1, 2018, obtained from the University of Bremen satellite sea ice product (seaice.uni-bremen.de, Spreen et al. (2008)).**






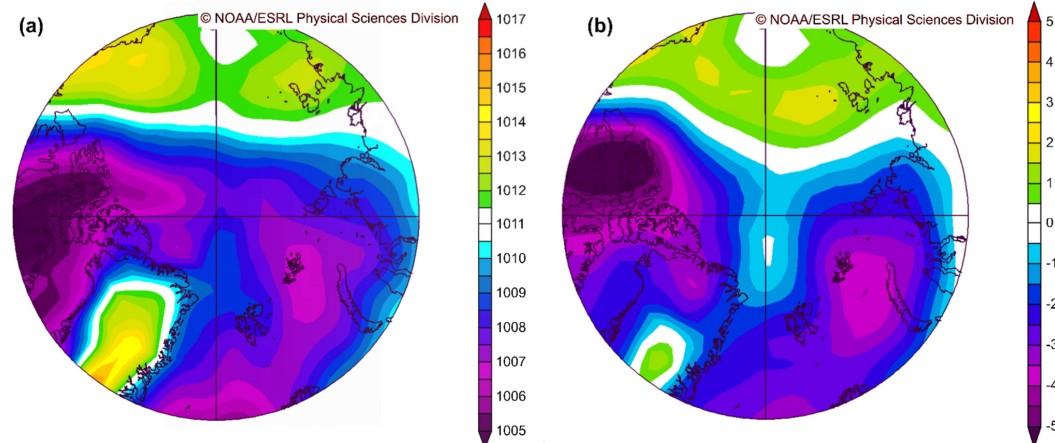

**Figure 2: Contour plots of (a) mean sea-level pressure and its (b) climatological anomaly (1981-2010) for the AO2018 measurement period. Image provided by the NOAA/ESRL Physical Sciences Division, Boulder Colorado from their Web site at http://www.esrl.noaa.gov/psd/. Based on NCEP Reanalysis data (Kalnay et al., 1996).**

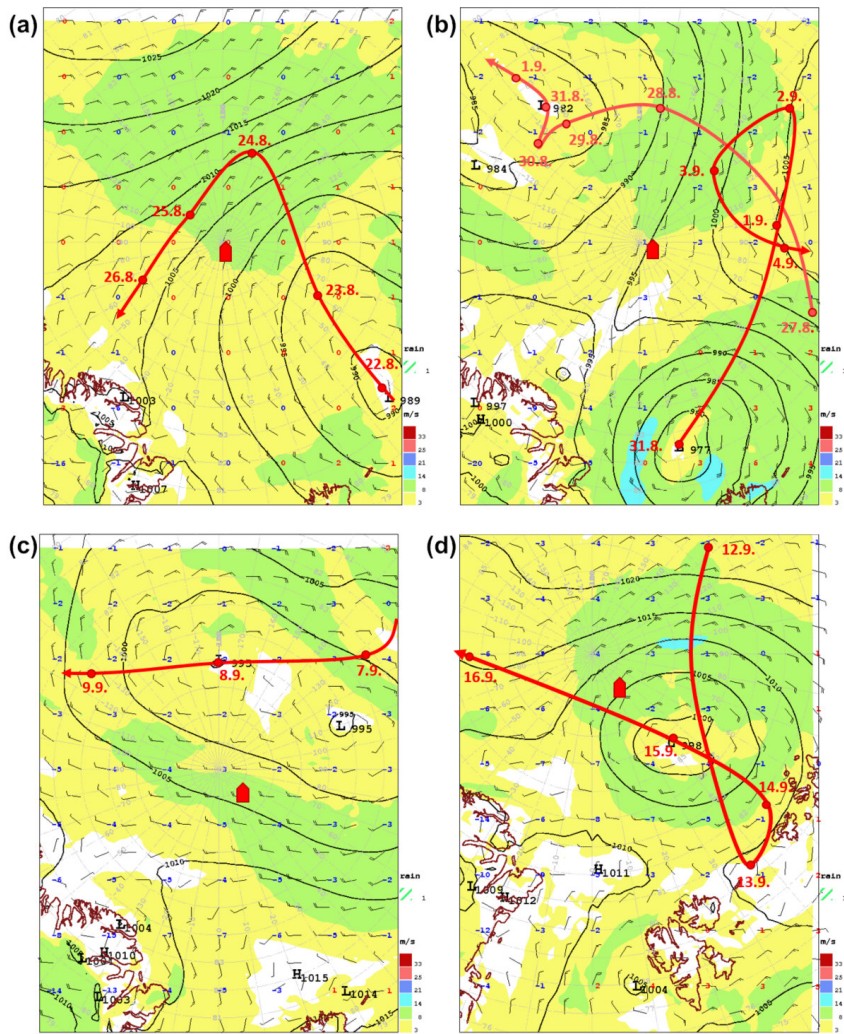

**Figure 3:** ECMWF charts showing sea-level pressure, near-surface wind and precipitation at 00:00 UTC for four days: (a) 22, (b) 31 August, (c) 08 and (d) 15 September, 2018. The figures also show storm tracks for the major low-pressure systems passing through the area with their low-pressure centres at 00:00 UTC on the respective day. The approximate location of the icebreaker Oden is marked by the red arrow.

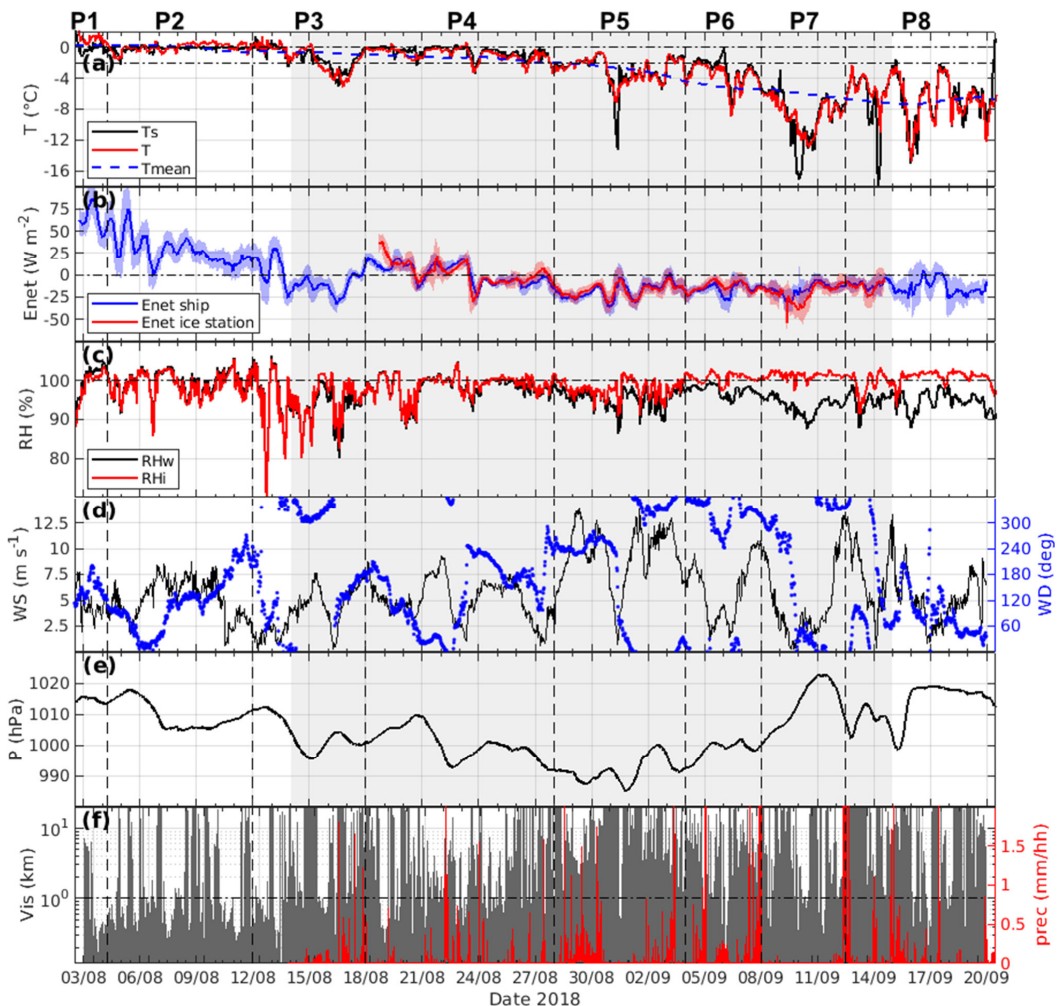

**Figure 4: Time series of (a) surface temperature, near-surface air temperature, and 14-day running mean of near-surface temperature, (b) 12-h running mean of surface energy ± one standard deviation (shaded area), (c) RHw and RHi, (d) wind speed and wind direction, (e) air pressure, (f) visibility and precipitation. Data in (a), (c), (e), (f) are from instruments installed on the 7th deck of the ship. Turbulent fluxes for calculation of (b) and wind measurements are from the foremast of the ship. Net radiation for calculation of (b) is from measurements on the ship (blue line) and the station on the ice floe (red line). Dashed lines mark the identified key periods, grey shaded area represents the drift period.**


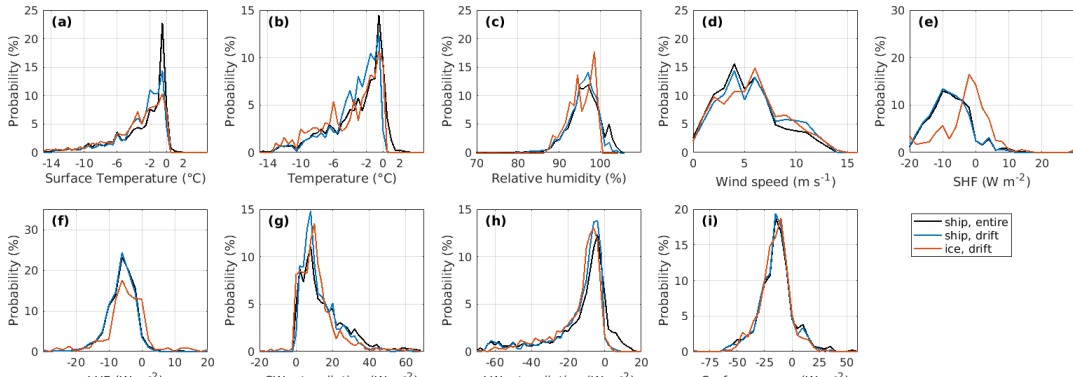

**Figure 5: Probability distributions of (a) surface temperature, (b) near-surface air temperature, (c) RHw, (d) wind speed, (e) sensible heat flux, (f) latent heat flux, (g) shortwave net radiation, (h) longwave net radiation, and (i) surface energy. Data are from measurements on board the ship for the whole measurement period (black lines), for the ice drift period only (blue line), and from the measurement station on the ice floe (orange line).**


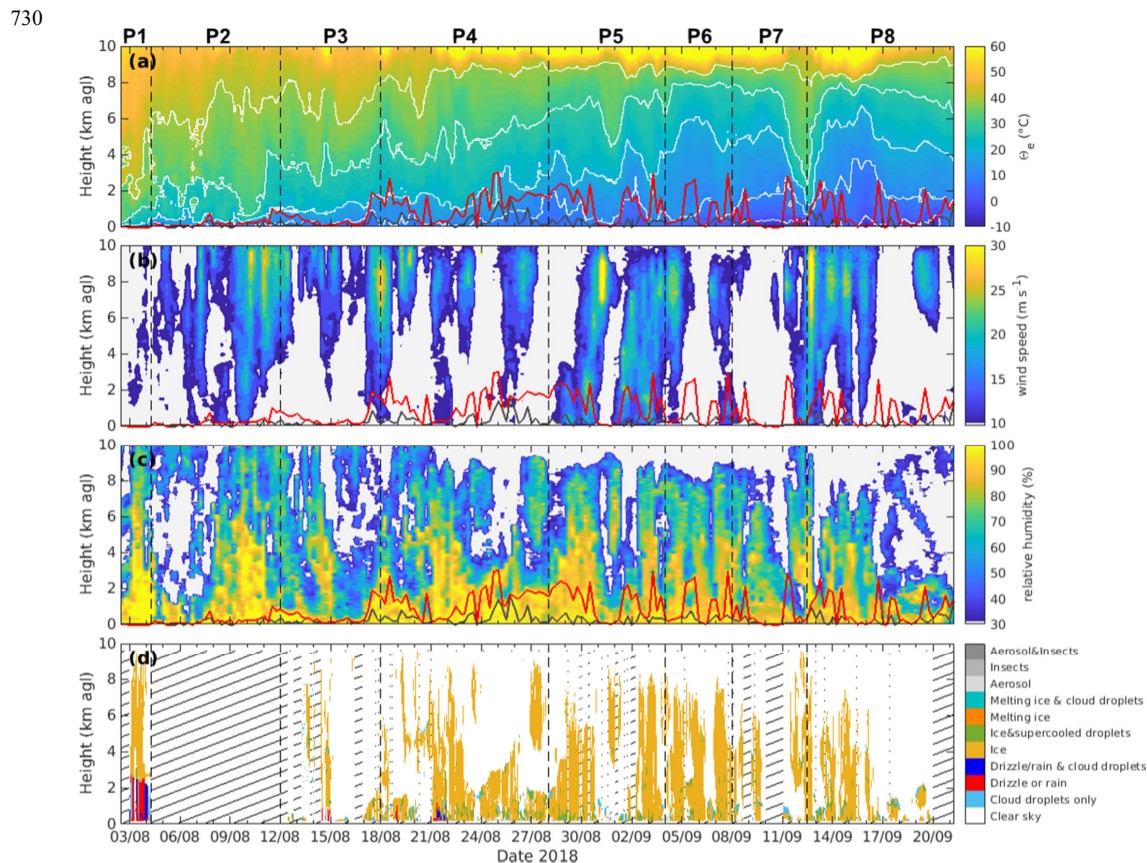

**Figure 6: Contour plots of (a) equivalent potential temperature, (b) wind speed, (c) RHw measured by radiosondes, and (d) cloud target classification from the Cloudnet algorithm. The red lines in (a)-(c) show the main inversion base height and the grey lines the surface mixed layer depth identified from radiosonde data. Dashed vertical lines mark the identified key periods.**

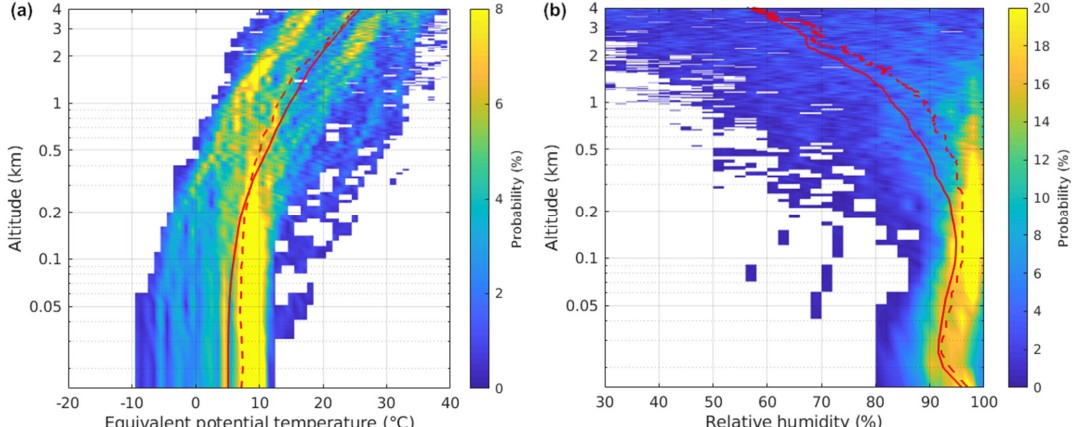

**Figure 7: Probability of (a) equivalent potential temperature and (b) RHw as a function of altitude. Note that the probability is calculated for each height; hence, for each layer the probability sums to 100%. The red solid line shows the mean and the red dashed line the median profile.**

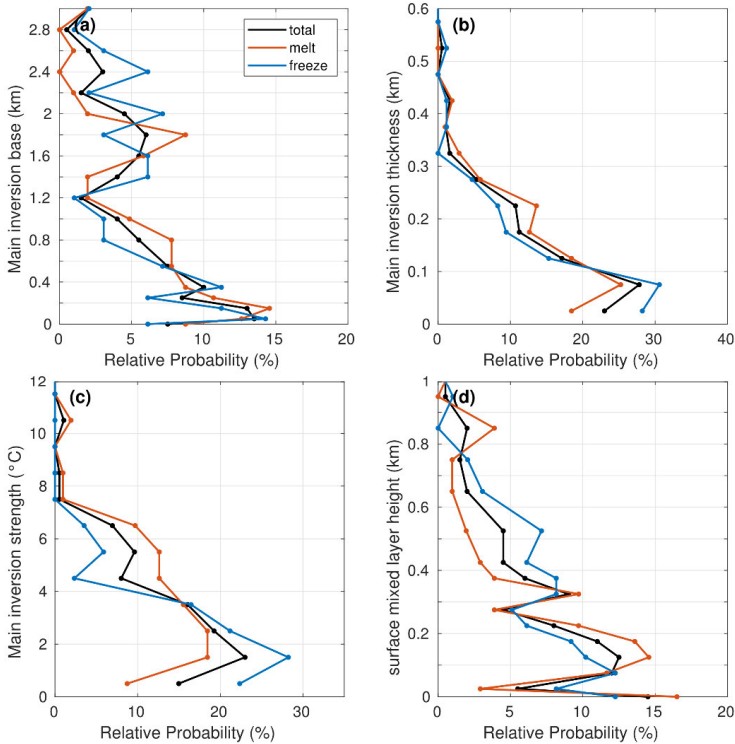

**Figure 8: Probability distributions of (a) main capping inversion base, (b) main capping inversion thickness, (c) main capping inversion strength, and (d) SML height, as detected from radiosonde data. Results are shown for the entire campaign (black lines), for the melt period before 28 August (orange lines) and for the freeze period of all data after the 28 August (blue lines).**






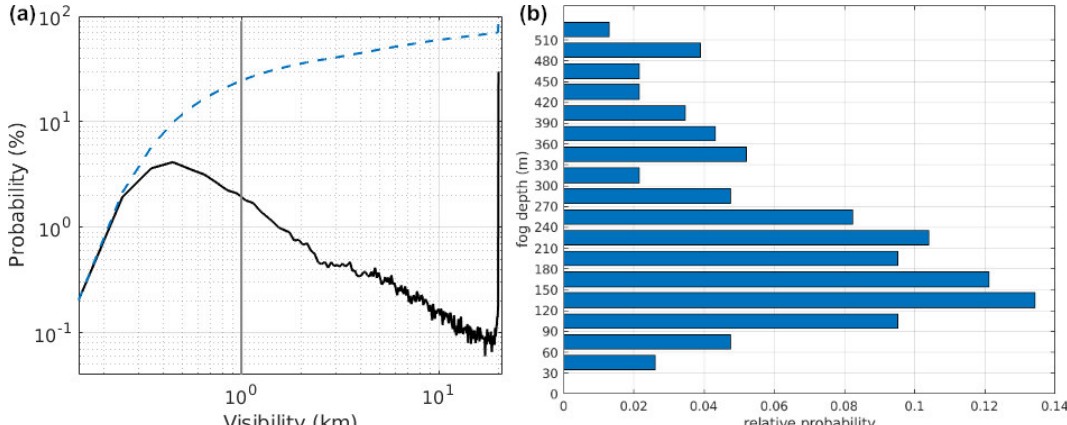

**Figure 9: (a) Probability distribution and accumulated probability (blue dashed line) of visibility. The black vertical line marks 1 km visibility. (b) Probability distribution of fog depths detected using radar RHI scans.**



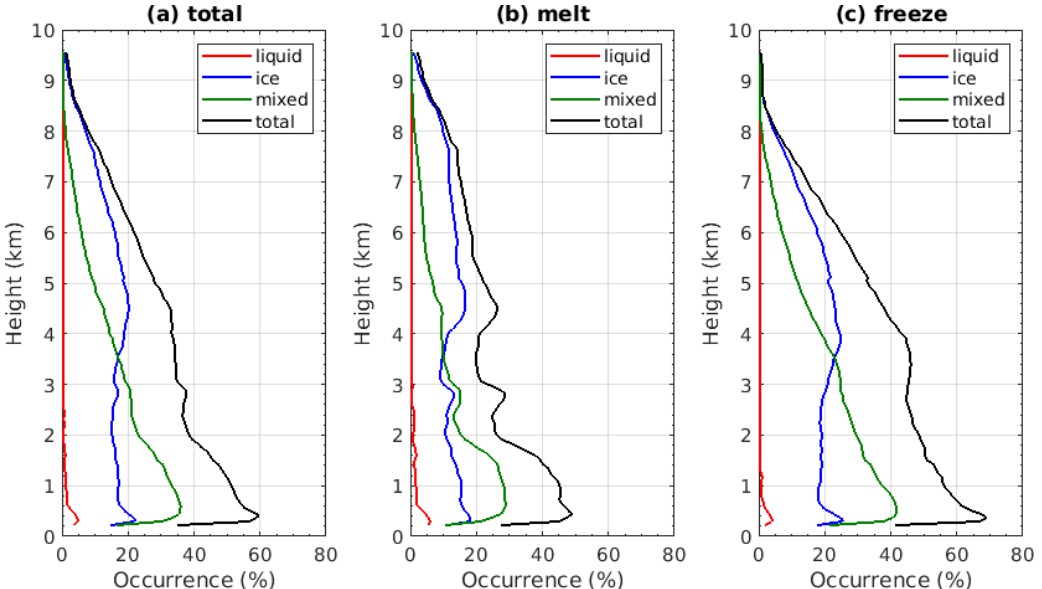

**Figure 10: Mean cloud occurrence per volume for different cloud types, obtained from the Cloudnet target classification product: (a) mean profiles for all available Cloudnet data, (b) for the melt period, and (c) for the freeze period.**






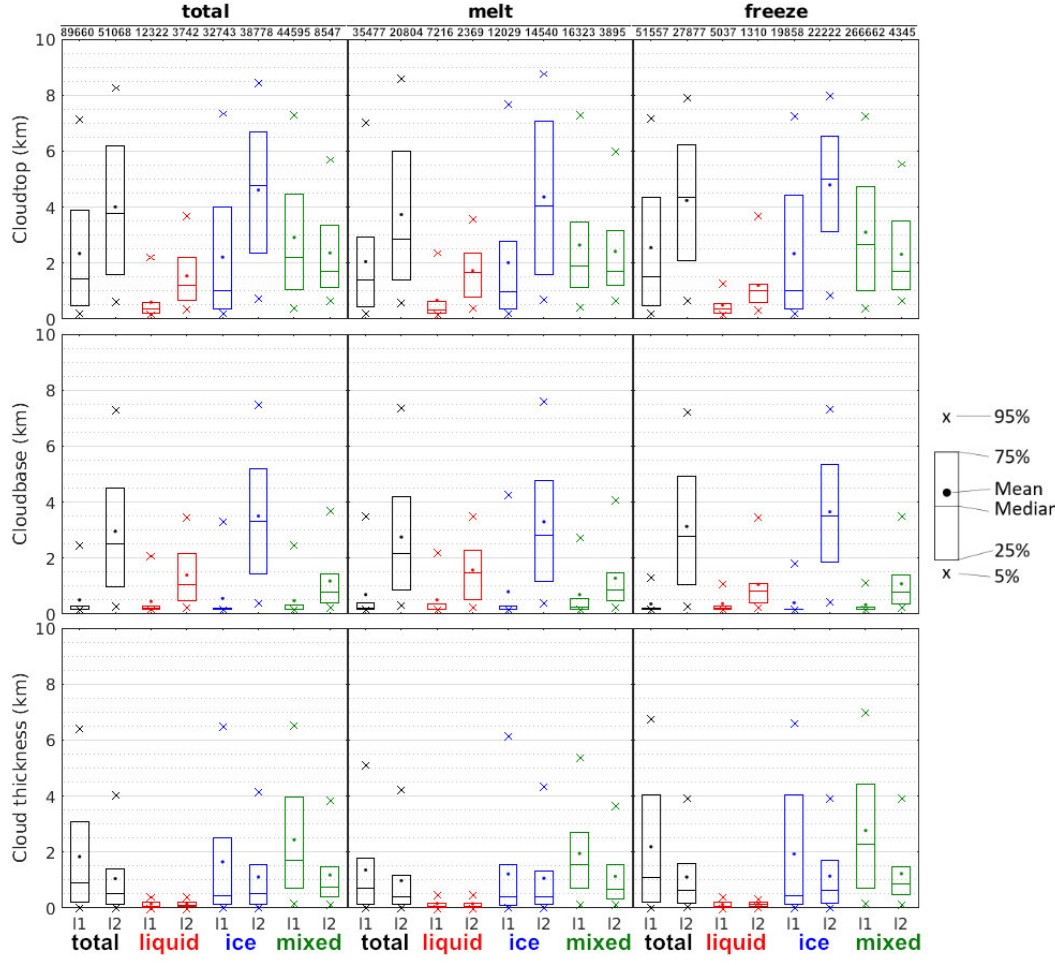

**Figure 11: Statistical overview of cloud top height (top row), cloud base height (middle row) and geometrical thickness (bottom row) for the entire AO2018 campaign (left column) as well as for melt (middle column) and freeze (right column) conditions. The colours indicate different cloud types. Results for the lowest two cloud layers (l1 and l2) are shown. The top row shows the number of profiles used for each boxplot.**





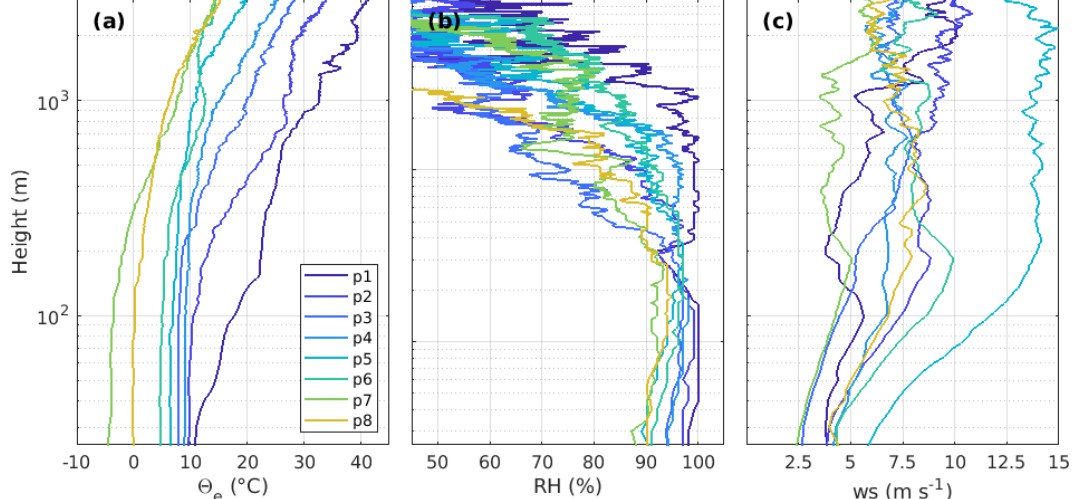

**Figure 12: Median profiles of (a) equivalent potential temperature, (b) RHw, and (c) wind speed for the eight key periods of the campaign.**



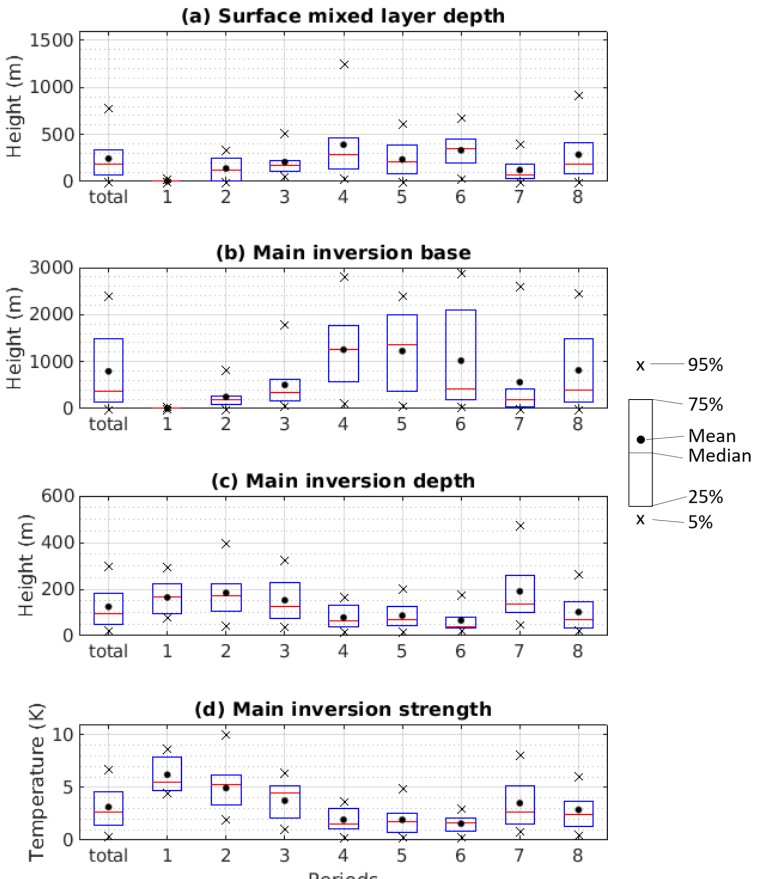

**Figure 13: Statistics on surface mixed layer depth, main inversion base height, main inversion depth, and strength for the whole campaign and for the 8 key periods.**




**Table 2: Percentage of coupled and decoupled boundary layer conditions for the whole campaign and for the 8 key periods. Furthermore, the relative amount of the decoupling type is given. Either by a weaker inversion below the main inversion or the BL is stable, meaning decoupling by a surface inversion.**

|  | total | P1 | P2 | P3 | P4 | P5 | P6 | P7 | P8 |
|---|---|---|---|---|---|---|---|---|---|
| coupled | 41 | 14 | 55 | 54 | 18 | 31 | 60 | 47 | 52 |
| decoupled | 59 | 86 | 45 | 46 | 82 | 69 | 40 | 53 | 48 |
| decoupled by weaker inversion | 77 | 0 | 36 | 100 | 94 | 80 | 100 | 80 | 69 |
| decoupled by surface inversion | 23 | 100 | 64 | 0 | 6 | 20 | 0 | 20 | 31 |

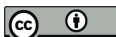



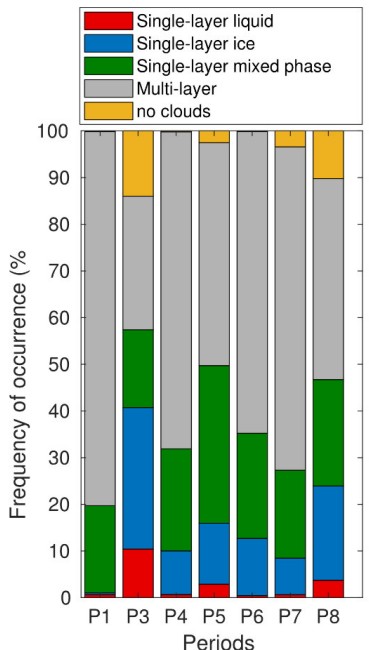

**Figure 14: Frequency of occurrence of different types of single-layer clouds, multilayer clouds and no clouds for all available key periods.**


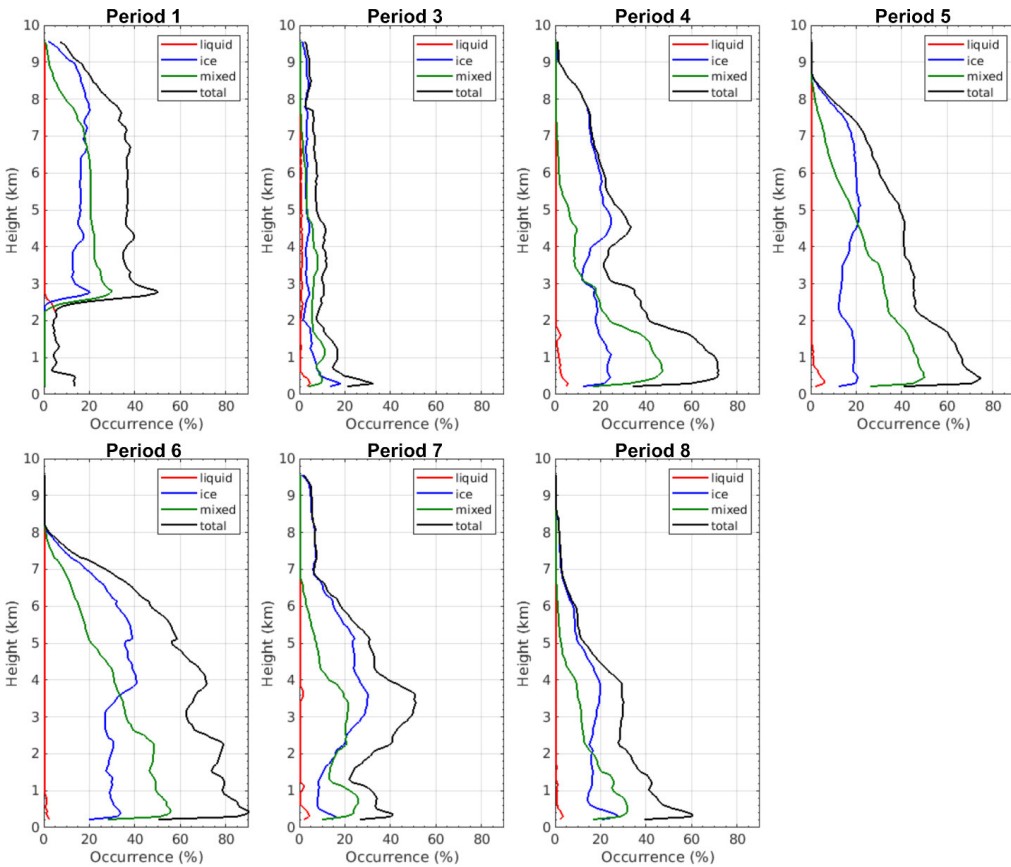

**Figure 15: Mean cloud occurrence per volume for different cloud types (colour coded), obtained from the Cloudnet target classification product for the eight key periods.**

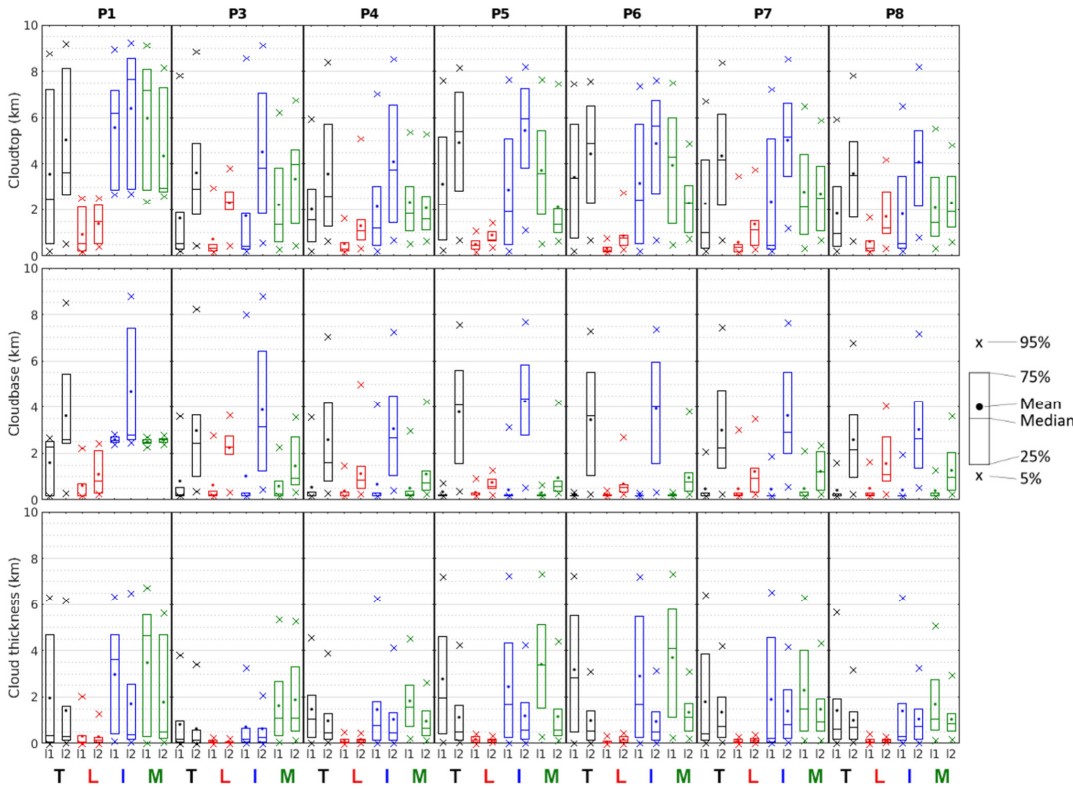

**Figure 16: Statistical overview of cloud top height (top row), cloud base height (middle row), and geometrical thickness (bottom row) for the eight key periods of AO2018. The colours indicate the different cloud types, total (T), liquid (L), ice (I), and mixed phase (M) clouds. Results for the lowest two cloud layers (l1 and l2) are shown.**






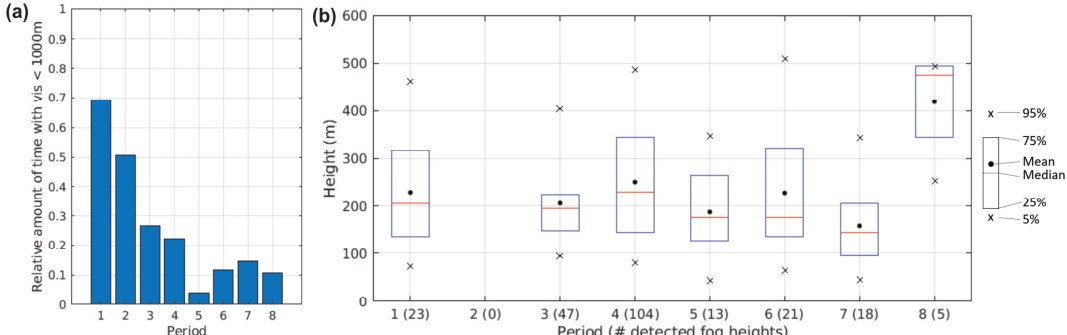

**Figure 17: Fog statistics. Relative amount of time with visibility <1000m for each period and statistics of fog depths for each period where data was available (period 2 no radar data available).**