# Peer review of "Meteorological and cloud conditions during the Arctic Ocean 2018 expedition"

_Atmospheric Chemistry and Physics, 2020_

## Referee Comment (RC1) · Anonymous Referee #1 · 19 May 2020

Full review of Vüllers et al., ACPD 2020

Summary of the manuscript acp-2020-219

The study titled "Meteorological and cloud conditions during the Arctic Ocean 2018 expedition" by J. Vüllers et al. gives an overview of the Arctic Ocean 2018 expedition with the Swedish icebreaker Oden into the Central Arctic in Aug-Sep 2018. It describes the atmospheric conditions encountered with an emphasis on the synoptic-scale events, a statistical analysis of the thermodynamic profiles of the atmosphere, the atmospheric boundary layer (ABL) characteristics, the cloud conditions and puts the AO2018 observations into context to previous expeditions into the Arctic Ocean. It was found that due to high cyclonic activity, the ABL was often well-mixed and a higher-than normal amount of multi-layer or mid-level clouds occurred while single-layer low-level stratocu-

mulus was observed less frequently than during previous expeditions. For several aspects of the statistical analysis like inversion strength/thickness/height as well as near-surface meteorological conditions, the observation period was divided into melt- and freeze up period as well as into synoptic-event dependent 8 periods. I would suggest the manuscript to be published after major revisions. The authors should address the following points: Major comments

l.242-247: Please connect this paragraph better to the previous results (and Fig6): under which synoptic and cloud conditions did which of the bimodal main capping inversion base heights occur? Under which synoptic and cloud conditions did which coupling state prevail?

- l.253: Explain why fog was misclassified as "Aerosol, Aerosol & Insects" by the Cloudnet algorithms. In Fig.31 d) these incidences were already filtered? Please add the fog depths in Fig.31 d). Griesche et al., 2019 (https://doi.org/10.5194/amt-2019-434) described a Raman-lidar based approach to introduce fog into the Cloudnet classification category based on Arctic cloud observations. – If possible, please apply this method to the HALO Photonics Doppler lidar/ceilometer observations and compare this method to your fog detection. As you mention, liquid occurrence would increase in Fig.10 if you included your fog detection results. In case you can use the lidar observations to detect fog, please include this fog category as liquid cloud in Fig 10 etc. to improve your cloud classification statistics. The lidar-based fog detection would also allow for fog characterization in P2 in Fig.17.

l.280f: A fixed threshold of Cloudnet mixed-phase layer depth of 700m seems a bit arbitrary for determination of single vs multiple cloud layers. The used reference Sotiropolou et al., 2014 is missing in the reference list. Based on your radar and radiosonde observations, you can determine if the clouds are thick mixed-phase clouds or actually multi-layer mixed-phase clouds as illustrated by Vassel et al., 2019 (https://doi.org/10.5194/acp-19-5111-2019). – Please apply this method to substantiate your AO2018 cloud statistics overview. This will then of course also affect the

single-vs-multi-layer cloud statistics in Fig.14+15 and the discussion on p.12ff.

Section 5.3: In multi-layer cloud conditions, please explain why you compare the statistics of first and second layer cloud base and depth (Fig.16)...if you do not do a more detailed/precise characterization of multi-layer clouds as suggested above, these differences don't seem to have a solid basis.

Precipitation conditions were only mentioned briefly in the manuscript even though cloud radar and micro rain radar measurements were made. Please describe Fig.4f more in detail and include a discussion of Fo of precipitation during the different periods, accumulated precip amount, snow vs rain vs supercooled drizzle.

Minor comments

- p.1 abstract: mention that the Arctic Ocean 2018 was a ship-based expedition with the Swedish icebreaker Oden - l.28f and following: citations to one specific fact should always be in chronological order - l.29: past five years is not true anymore since we are now in 2020, please rephrase - l.45: mid-latitudes are only well-characterized in the Northern Hemisphere, please adjust. - l.52: radiative effect of mixed-phase clouds is surface dependent: over ice warming, over open ocean not. Please be more specific. - l.66f: The comparability of AO2018 to the mentioned four previous campaigns would benefit from a map and a table: A map showing the focus areas of the other campaigns and the corresponding sea ice cover (in comparison to Fig1.) and a table stating the time periods of the other campaigns - did they all happen during Aug-Sep or earlier in the summer? - l.86: add websites of the two projects as footnote - l.89: "atmospheric" remote sensing instruments - l. 90f: Why was only the lidar horizontally stabilized? - l.96: Just to clarify, 6 hourly radiosonde data instead of GDAS or ECMWF re-analysis was used in the Cloudnet processing of the data? - p.4: Please include a photo of the Oden and label the positions of the mentioned long-list of instruments. - l.102 and 106: How did the two ceilometers compare? - l.127: Doesn't the LI-COR LI-7500 measures fluxes of water vapor and CO2? - l.128: at which altitude above the snow surface

were the solar- and IR radiometers installed? Please add. - l.161: Define how the "net surface energy" was determined, it should be the sum of the net radiation + latent + turbulent heat fluxes + ground (soil) heat flux. – Did you measure ground heat flux? Or do you refer to "net surface radiation" instead of energy? (Also Fig5f) Please be precise in your use of "net surface radiation budget" vs "net surface energy budget" throughout the manuscript - l.188: "near-surface air" instead of "surface" - l.200: "wind speeds" - l.253: You have not mentioned the duty cycle of the radar measurements previously. In the measurement description please add the cycle of cloud radar RHI scans (1x/hour? 2x/hour?...?) as well as the RHI scan angles. - l.274: According to Fig10 b) and c), mixed-phase clouds had a higher occurrence frequency than ice clouds below 3.5km for both, the melt and freeze-up period. Also, mention that the Cloudnet target classification likely underestimates the FoO of mixed-phase clouds during multi-layer situations since it only classifies liquid in the presence of a lidar signal which in turn gets fully attenuated at an optical thickness of three though. - l.287: I suppose the listed liquid cloud statistics again do NOT include the fog classification? – If so, please update after including the fog occurrence as liquid cloud. -l.290: It sounds like as if you used the Cloudnet target classification mask for cloud base altitude determination. Why don't you use the multiple lidar observations (HALO, 2x ceilometer) to determine cloud base? What are the lowest observation range gates of the lidars? - l.291: Again, emphasize that ice clouds refer to "ice clouds and ice precipitation" leading to such deep cloud depths. - l.325: Here you state that P7 was characterized by mostly cloud-free conditions. Fig.14 however shows that during P7 cloud-free conditions occurred only for about 5% of the time while for P3 and P8 cloud-free conditions occurred more frequently, namely about 15%, and 10% of the time. – Please adjust. - l.404: Here you mention that cloud radar data was only available during the first 3 days of P8. – Please bundle data gaps earlier on in one paragraph when describing the measurements.

Comments on Figures:

Fig. 4: Please indicate the phase of the precipitation reaching the ground (snow/rain

or ice/mixed-phase/liquid) in panel e). Did you experience supercooled drizzle?

Fig.5: I am surprised that the PDF of SW/LW/Net surface radiation balance do not differ much for the ship- vs. ice period. – Can you explain why?

Fig.6: What do the grey stripes in d) represent? – Ok, seems like the answer is on p.10: No radar observations (and thus Cloudnet target classifications) were possible during the ice-breaking period between Aug 4-12. This should be mentioned much earlier. Also in Fig. 14 it should be explicitly stated in the caption that P2 had to be excluded.

---

## Referee Comment (RC2) · Anonymous Referee #2 · 24 Jul 2020

This manuscript describes shipborne and ice pack observations taken during the AO2018 expedition in the Arctic Ocean. High-level motivating factors for this deployment and instrumentation are described. General results from the campaign are also provided that summarize overall atmospheric state, near-surface energy exchanges, and cloud-related properties (composition, occurrence, and macrophysical properties). Special attention is given to low-level thermodynamic properties that are especially important in the Arctic environment. Gross comparisons between the current study and previous Arctic observations are also provided for necessary context.

Overall, the manuscript provides valuable information that should appeal to a wide range of Arctic-centric researchers. It nicely describes the motivation for undertaking the AO2018 campaigns and conveys high-level results that are interesting and com-

pelling. The topic is also relevant for ACP. I encourage the authors to consider addressing the mostly minor clarifying issues outlined below. I look forward to seeing this manuscript published in ACP.

Note: I reviewed this manuscript with the expectation that its purpose is to first and foremost advertise the AO2018 field campaign. Its secondary purpose is to provide high-level, general results that illustrate the utility of the observations. I do not expect every outstanding Arctic-related scientific puzzle to be answered within this manuscript, but hope that it properly sets the stage for future research using this dataset and future observational campaigns.

********************* Specific Comments **********************

Introduction: Nicely done. The introduction properly motivates the AO2018 deployment within the broader context of Arctic research issues that still need to be addressed by the community. It is succinct, yet manages to provide ample background information. Reviews too often focus on items that should be changed or clarified, so I wanted to take the opportunity to express positive feedback regarding the introduction.

Lines 164-165: How is shortwave albedo quantitatively estimated using imagery?

Line 168 and Fig. 4: How is near-surface temperature defined? Is this 2-meter temperature, or some other level?

Sections 4.3 and 4.4: A few sentences could benefit from simple restructuring or key comma insertions to reduce run-on sentences. Similar minor issues also appeared in other sections, but can also be rectified during final editing processes. I wanted to mention them here, though, since I found them marginally distracting. A few examples are provided below.

Lines 232-233: "If no temperature inversion could be identified, the strongest stable layer..."

Lines 239-240: "...to the lowest measurement heights of the radiosonde (30m), it was

classified. . ."

Lines 257-258: A general question regarding Cloudnet and radars employed in this campaign. The authors mention that Cloudnet does not observe 49% of fog events since the first usable radar observation is about 150m above ground level (i.e., many fog events are very shallow). Does radar sensitivity also affect fog detection statistics? Stated another way, it would be nice for the authors to advertise instrument sensitivity somewhere in the manuscript (e.g., minimum radar detectable signal).

General Cloudnet question: I assume Cloudnet algorithms are only applied to scanning radar observations and not the Micro Rain Radar (MRR)? Depending on how the MRR is configured, it can provide valuable observations below 150 m. But MRR sensitivity probably will not detect fog and non-precipitating clouds.

Section 4.4: How are cloud phases determined? Combined radar-lidar observations? Radar only?

Lines 332-334 and a few other locations throughout the manuscript: I suggest economizing wording and removing nominalisations to improve readability. I definitely do not want to completely alter the authors' voice, but simple changes like the following will be impactful:

"Equivalent potential temperature profiles are strongly stratified in the lowest 150 m, . . .."

Section 6: This is an important section. I found myself begging for comparisons to previous research when results were presented in earlier sections. In hindsight, though, I like how the authors refrained from comparing to previous studies until this discussion-like section. It serves as a nice overall summary that effectively complements the conclusions.

Conclusion: I like how the last paragraph begins, but I somehow feel that it ends in a disappointing fashion. It feels. . ..incomplete? I am not sure how to properly describe

it. Maybe ending with a strong statement about how observations can help answer the lingering question of increasing Arctic cyclone activity shown by reanalyses? Simply swapping the second and third sentences might help, with an appropriate bridge that connects the first and second sentences. Something like "For instance, reanalysis data indicates an increase......". Then relate the observations from the current study and need for continued observations to definitively answer this outstanding scientific question. The manuscript will not suffer tremendously if the last paragraph is not altered, but I encourage the authors to concoct a more impactful ending paragraph.

Fig. 2: Suggest adding [hPA] units to colorbars or figure caption.

Fig. 3: Wind speed colorbar labels quite difficult to discern. Are the blue and red numbers located at regular grid intervals the surface temperature?

Fig. 4: Suggest adding "vertical dashed lines" to the last sentence of the caption to distinguish between horizontal dash-dot lines in a few of the figure panels. Also consider adding "(e.g., P1, P2, ..., P8)" to the figure caption to explicitly advertise that these labels are associated with respective observational periods.

General Question: Is there any reason why precipitation statistics were not shown? I completely understand the need to draw a proverbial analysis line somewhere – every detail cannot be shown. But I am curious if precipitation statistics have been analysed or plan to be analysed in a separate study. At the very least, this topic could be added to the future research discussion in the conclusion.

General Question: Did this campaign encounter any specific measurement complications (e.g., instrument performance, logistics, etc.)? If yes, it would be great to briefly describe some of them to both advertise how difficult it is to operate shipborne instrumentation thousands of kilometers from population centers in a rather hostile environment and serve as valuable feedback for other researches that might consider adopting similar measurement techniques in future field campaigns.

---

## Author Comment (AC1) · 23 Oct 2020

**We would like to thank the Reviewers for the comments and suggestions that helped to improve the quality of our publication. Reviewer comments are given in black. Our replies are given in red and new text in the manuscript in blue.**

**Reply to Reviewer 1**

**MAJOR COMMENTS**

l.242-247: Please connect this paragraph better to the previous results (and Fig6): under which synoptic and cloud conditions did which of the bimodal main capping inversion base heights occur? Under which synoptic and cloud conditions did which coupling state prevail?

We added some information to the paragraph. A detailed discussion of the temporal evolution of the inversion characteristics and cloud conditions is already provided in section 5.2 and shows that surface inversions mainly occur at the beginning of the campaign whereas the higher main inversions are connected to frontal systems. New text in section 4.3:

*Surface inversions occurred preliminary during the calm conditions, at the beginning of the campaign (Fig.7). The probability distributions of the capping inversion and SML characteristics are shown in Fig. 9. The main capping inversion base height shows a bimodality with a maximum below 400 m and another one around 1500 m. High main capping inversions are mostly connected to the passage of frontal systems.*

l.253: Explain why fog was misclassified as "Aerosol,Aerosol&Insects" by the Cloudnet algorithms.
  In Fig.31 d) these incidences were already filtered?
  Please add the fog depths in Fig.31 d).
  Griesche et al., 2019 (https://doi.org/10.5194/amt-2019-434) described a Raman-lidar based approach to introduce fog into the Cloudnet classification category based on Arctic cloud observations.
  If possible, please apply this method to the HALO Photonics Doppler lidar/ceilometer observations and compare this method to your fog detection.
   As you mention, liquid occurrence would increase in Fig.10 if you included your fog detection results. In case you can use the lidar observations to detect fog, please include this fog category as liquid cloud in Fig 10 etc. to improve your cloud classification statistics. The lidar-based fog detection would also allow for fog characterization in P2 in Fig.17.

Cloudnet was developed and tuned for mid-latitude conditions. Cloudnet does not have a separate fog class. Height bins with a low lidar signal and no radar signal are considered to be either aerosols or aerosols and insects, which fits for mid-latitudes.
In the very clear Arctic atmosphere, the boundary layer has exceptionally low aerosol concentrations, and hence fog typically has low droplet number concentrations and the combination of radar backscatter & Doppler velocity on which the classification is based, best matches that for aerosol/insects in mid-latitudes.
During ACSE we saw an abundance of Aerosols as well as Aerosols & Insects signals in the lowermost height levels of the Cloudnet target classification. This seemed unreasonable so we cross-checked these time periods with coincident measurement with a visibility sensor (as well as camera images). It turned out that periods which were classified as Aerosol and Aerosol & Insects close to the surface correspond well to fog occurrence.
Using the additional information from the visibility sensor, we introduced the fog classification for the analysis of the ACSE measurements and adapted the method to the

data collected during AO2018.
A fog classification as part of the Cloudnet retrieval is still under development under the lead of Ewan O'Connor.

Data in Fig 4d was not filtered. It shows the original Cloudnet categories. Filtering is applied prior to all subsequent analysis.

Using the fog detection method of Griesche would give us information of fog occurrence However, it might not necessarily give fog heights as the HALO lidar signal might already be completely attenuated before reaching the top of the fog layer in thick fog (Griesche et al had a different lidar). Hence, the obtained data would only give us fog occurrence, which we already get from the visibility sensor. Therefore, we won't be able to use their approach or include the results in the mentioned figures & results. We tried to add fog depth to Fig. 6d (Now 7d) but decided against it because the y-axis goes up to 10km so fog depths of 150m are not properly visible in the figure.

*l.280f: A fixed threshold of Cloudnet mixed-phase layer depth of 700m seems a bit arbitrary for determination of single vs multiple cloud layers. The used reference Sotiropolou et al., 2014 is missing in the reference list. Based on your radar and radiosonde observations, you can determine if the clouds are thick mixed-phase clouds or actually multi-layer mixed-phase clouds as illustrated by Vassel et al., 2019 (https://doi.org/10.5194/acp-19-5111-2019). – Please apply this method to substantiate your AO2018 cloud statistics overview. This will then of course also affect the single-vs-multi-layer cloud statistics in Fig.14+15 and the discussion on p.12ff.*

We added the reference to Sotiropolou et al. 2014.
Furthermore, we applied Vassel et al's method to our data set. This gives us additional information of potential seeding MLCs for the drift period, when both radiosondes & cloud radar data were available. We added the results in section 4.3 and 5.3. They show that our hypothesis is plausible. The data show that there are potentially 48% of the time multi-layer seeding clouds if we assume an ice crystal size of 400µm.  The results in Fig. 14+15 (now 15+16) remain unchanged as Vassels approach cannot directly be applied to our results as the required radiosondes have a time resolution of 6 hours, whereas Cloudnet has a 30sec time resolution. The substantial variability on time scales much shorter than 6 hours means we cannot simply interpolate the radiosonde saturation measurements to the Cloudnet time base. However, we have added the results from Vassels method as a separate figure for the individual periods to Fig. 14 (now Fig. 15) and added it to the discussion of the results in section 5.3.
The overall description of the applied method was added to section 4.4:
*Vassel et al. (2019) provide a method to detect possible seeding events combining radiosonde and radar data. In step one ice-supersaturated and ice-subsaturated layers are identified using relative humidity data from radiosonde profiles. The sublimation of an ice crystal through the subsaturated layer is calculated assuming an initial size of 400 µm, but is also calculated for 100 and 200 µm. If the ice crystal is not fully sublimated when reaching a lower supersaturated layer, potential seeding is taking place. In a second step the results are cross-checked for actual cloud occurrence using radar reflectivity. Radiosonde data were available every 6 hours and radar data were only available during the drift period. Hence, Vassel et al.'s method was applied 6 hourly between 13 August until 14 September. 12% of the data show no cloud occurrence. Results for single layer and multi-layer clouds vary with assumed ice crystal size. Single-layer clouds occur in 32% (r=100 µm: 50%, r=200 µm: 38%) of the analysed profiles for an assumed size of 400 µm.*

*Non-seeding multi-layer clouds occur in 13% (r=100 µm: 8%, r=200 µm: 11%), seeding multi-layer clouds in 37% (r=100 µm: 18%, r=200 µm: 30%) of the profiles and profiles with both, seeding and non-seeding layers in 11% (r=100 µm: 7%, r=200 µm: 9%). These results strengthen our hypothesis of multi-layer seeding clouds and these limitations should be kept in mind for comparisons with other observational results not obtained with the Cloudnet algorithm.*

[Figure]

Figure 15: (a) Frequency of occurrence of different types of single-layer clouds (SLC), multi-layer clouds (MLC) and no clouds from the Cloudnet results for all available key periods. P2 was excluded from the analysis as no radar data were available. (b) Frequency of occurrence of no clouds, SLC and potentially seeding and non-seeding MLC using Vassel et al.'s (2019) method for assumed ice crystal sizes of 400 µm (left), 200 µm (middle) and 100 µm (right).

*Section5.3: In multi-layer cloud conditions, please explain why you compare the statistics of first and second layer cloud base and depth (Fig.16)..if you do not do a more detailed/precise characterization of multi-layer clouds as suggested above, these differences don't seem to have a solid basis.*

We show the statistics of the first two layers separately as merging the results of the cloud layers would skew the statistics. We decided to only show results from the first two layers as they seem the most relevant for boundary-layer cloud interactions. This analysis also follows the same approach as that of previous studies, and thus provides a consistent means of comparing conditions across studies.

*Precipitation conditions were only mentioned briefly in the manuscript even though cloud radar and micro rain radar measurements were made. Please describe Fig.4f more in detail and include a discussion of Fo of precipitation during the different periods, accumulated precip amount, snow vs rain vs supercooled drizzle.*

We have replaced the precipitation intensity in Fig.4 f (now Fig. 5) with the accumulated precipitation amount. Furthermore, we added form of precipitation to Fig. 5 and to the discussion. The measurements show that almost all precipitation fell as snow or ice (93.3%), another 4.6% as freezing drizzle and 1.8% as freezing rain. Only 1.3% of the precipitation was liquid, either drizzle or rain. Larger precipitation events were linked to frontal systems.

[Figure]

Figure 5: Time series of (a) surface temperature, near-surface air temperature, and 14-day running mean of near-surface temperature, (b) 12-h running mean of surface energy ± one standard deviation (shaded area), (c) RHw and RHi, (d) wind speed and wind direction, (e) air pressure, (f) visibility, accumulated precipitation and precipitation type.   Data in (a), (c), (e), (f) are from instruments installed on the 7th deck of the ship. Turbulent fluxes for calculation of (b) and wind measurements are from the foremast of the ship. Net radiation for calculation of (b) is from measurements on the ship (blue line) and the station on the ice floe (red line). Vertical dashed lines mark the identified key periods P1 to P8 and the grey shaded area represents the drift period. The precipitation type in (f) is color coded. Grey is missing data, blue is drizzle and rain, green is snow and ice.

**MINOR COMMENTS**

- *p.1 abstract: mention that the Arctic Ocean 2018 was a ship-based expedition with the Swedish icebreaker Oden*
  Done
  *The Arctic Ocean 2018 (AO2018) took place in the central Arctic Ocean in August and September 2018 on the Swedish icebreaker Oden*

- *l.28f and following: citations to one specific fact should always be in chronological order*
  Changed order of citations to chronological order

- *l.29: past five years is not true anymore since we are now in 2020, please rephrase*
  Updated the statement with data from latest Arctic Report
  *The past six years (2014-2019) were the warmest in the record starting in 1900 (Richter-Menge et al., 2019).*

- *-l.45: mid-latitudes are only well-characterized in the Northern Hemisphere, please adjust.*
  Done
  *Cloud feedback processes in the Arctic are particularly challenging for models as there are notable differences to the more commonly studied lower latitudes and tropics*

- *l.52: radiative effect of mixed-phase clouds is surface dependent: over ice warming, over open ocean not. Please be more specific.*
  changed it to ice surface.

- *l.66f: The comparability of AO2018 to the mentioned four previous campaigns would benefit from a map and a table: A map showing the focus areas of the other campaigns and the corresponding sea ice cover (in comparison to Fig1.) and a table stating the time periods of the other campaigns - did they all happen during Aug-Sep or earlier in the summer?*
  A map showing the other cruise tracks and a table with the expeditions times was added to the manuscript. See new Fig. 1 and Table 1

- *l.86: add websites of the two projects as footnote*
  Done

- *l.89: "atmospheric" remote sensing instruments*
  Done

- *l. 90f: Why was only the lidar horizontally stabilized?*
  Since all the primary measurements used here are made during the ice drift, when Oden is moored to the floe, none of the instruments really need to be stabilised against ship motion. We stabilised the lidar so that it could usefully continue to make Doppler velocity measurements during transit, but only because we happen to have a motion-stabilised platform for it, made for the ACSE cruise where all the measurements were made underway. The radar was not routinely operated underway because of the risk of damage to the bearings on the steerable antenna during the harsh vibration of icebreaking. None of the other measurements require motion stabilisation here.

- *l.96: Just to clarify, 6 hourly radiosonde data instead of GDAS or ECMWF re-analysis was used in the Cloudnet processing of the data?*
  Yes, radiosonde data was used for Cloudnet processing

- *p.4: Please include a photo of the Oden and label the positions of the mentioned long-list of instruments.*
  We added the requested photo of Oden with instrument locations (new Fig. 2)

- *l.102 and 106: How did the two ceilometers compare?*
  There is a mean bias between the Vaisala and Campbell units of ~50m, but changes over time track closely between the two systems. The mean offset is most likely a result of how cloud base (threshold) is defined in the individual instrument's internal processing algorithms

- *l.127: Doesn't the LI-COR LI-7500 measures fluxes of water vapor and CO2?*

In principle, yes the LI-7500 measured high rate fluctuations of $CO_2$ and is widely used to measure $CO_2$ fluxes. However, there are many problems with the instrument (contamination of $CO_2$ signal by overlap with water vapour absorption band, motion (and indeed orientation) sensitivity of $CO_2$ signal,…). These problems are particularly acute at sea (high humidity flux) and on ships (high degree of motion), and no one in the air-sea flux community trusts the instrument for $CO_2$ fluxes made from ships. $CO_2$ fluxes measurements were made from the foremast turbulence system, but using a more sensitive instrument, a Los Gatos Research Fast Greenhouse Gas Analyzer (FGGA). Measurements from a previous cruise with the same instruments are published in Prytherch et al. (2017). A second set of $CO_2$ flux measurements were made at the open lead site using a closed path Licor 7200 sensor. These have been submitted separately for publication.

*Prytherch, J., I. M. Brooks, P. Crill, B. Thornton, D. J. Salisbury, M. Tjernström, L. Anderson, M. C. Geibel, C. Humborg, 2017: Direct determinatio Prytherch, J., I. M. Brooks, P. Crill, B. Thornton, D. J. Salisbury, M. Tjernström, L. Anderson, M. C. Geibel, C. Humborg, 2017: Direct determination of the air-sea CO2 gas transfer velocity in Arctic sea ice regions, Geophys. Res. Letts, 44, doi:10.1002/2017GL073593n of the air-sea CO2 gas transfer velocity in Arctic sea ice regions, Geophys. Res. Letts, 44, doi:10.1002/2017GL073593*

*- l.128: at which altitude above the snow surface were the solar- and IR radiometers installed? Please add.*

   At 1.5 m height. Added to text.

*- l.161: Define how the "net surface energy" was determined, it should be the sum of the net radiation + latent + turbulent heat fluxes + ground (soil) heat flux.*

   Net surface energy is defined as Enet= net radiation – lhf – shf; we were not trying to define the total surface energy budget but only the atmospheric side of the energy budget

*– Did you measure ground heat flux? Or do you refer to "net surface radiation" instead of energy? (Also Fig5f) Please be precise in your use of "net surface radiation budget" vs "net surface energy budget" throughout the manuscript*

   No we didn't as we focused on clouds we measured atmospheric side of the energy budget only (radiation + turbulent fluxes). We went over the manuscript and tried to make sure that descriptions are precise when referring to radiation or energy budgets.

*- l.188: "near-surface air" instead of "surface"*

   Done

*- l.200: "wind speeds"*

   Done

*-l.253: You have not mentioned the duty cycle of the radar measurements previously. In the measurement description please add the cycle of cloud radar RHI scans (1x/hour? 2x/hour?...?) as well as the RHI scan angles.*

   Duty cycle was an RHI scan every 30 minutes, staring vertically the rest of the time. It was added to the text.

*- l.274: According to Fig10 b) and c), mixed-phase clouds had a higher occurrence frequency than ice clouds below 3.5km for both, the melt and freeze-up period. Also, mention that the Cloudnet target classification likely underestimates the FoO of mixed-phase clouds during multi-layer situations since it only classifies liquid in the presence of a lidar signal which in turn gets fully attenuated at an optical thickness of three though.*

Thank you for the comment. We corrected the sentence and added the potential uncertainty due to lidar attenuation to the text.

*These secondary maxima reflect the frequent occurrence of multiple cloud layers during AO2018. Mixed phase clouds were the most abundant cloud type, occurring below 3.5 km and some mixed phase clouds were observed up to a height of 8 to 9 km. Above these levels ice clouds dominated. However, mixed phase clouds might be underestimated in multi-layer cloud situation, if the lidar signal gets fully attenuated.*

*- l.287: I suppose the listed liquid cloud statistics again do NOT include the fog classification? – If so, please update after including the fog occurrence as liquid cloud.*

Fog is included as liquid cloud if it is reaching the lowest radar range gate at 156 m.

*-l.290: It sounds like as if you used the Cloudnet target classification mask for cloud base altitude determination. Why don't you use the multiple lidar observations (HALO, 2x ceilometer) to determine cloud base? What are the lowest observation range gates of the lidars?*

We decided against using the ceilometer results as we would like to keep the results comparable to other campaigns where the Cloudnet algorithm is applied, and on the consistent cloudnet time/height grid. Note, however, that the cloudnet cloud base is derived from the lidar/ceilometer measurements, filtered through the algorithm. The additional ceilometer measurements that we have might not be available for other campaigns. Since we have already seen a 50m mean bias between the two ceilometers, resulting from different internal processing algorithms/thresholds, these do not necessarily provide a 'better' measure of cloud base than that from cloudnet.

*- l.291: Again, emphasize that ice clouds refer to "ice clouds and ice precipitation" leading to such deep cloud depths.*

Added to manuscript

*- l.325: Here you state that P7 was characterized by mostly cloudfree conditions. Fig.14 however shows that during P7 cloud-free conditions occurred only for about 5% of the time while for P3 and P8 cloud-free conditions occurred more frequently, namely about 15%, and 10% of the time. – Please adjust.*

Thanks for pointing that out. We changed the sentence accordingly.

*- l.404: Here you mention that cloud radar data was only available during the first 3 days of P8. – Please bundle data gaps earlier on in one paragraph when describing the measurements.*

We added a description of data availability, which is shown in Table 2, to section 3 of the manuscript

*Ship based instrument systems were operated nearly continuously throughout the whole expedition (Table 2). The scanning doppler cloud radar could not be operated during heavy ice breaking between 5 and 13 August due to excessive vibration. The radar performed one Range-Height-Indicator (RHI) scan every 30 min and was operated in vertical stare mode the rest of the time. On the transit out of the ice it operated in vertical stare mode only. Precipitation data from the present weather sensor are only available from 13 August onwards. The rest of the data sets only have smaller data gaps.*

**Comments on Figures**:

Fig. 4: Please indicate the phase of the precipitation reaching the ground (snow/rain or ice/mixed-phase/liquid) in panel e). Did you experience supercooled drizzle?

We added precipitation type to Fig. 4f (now fig. 5f shown above). The measurements show that almost all precipitation fell as snow or ice (93.3%), another 4.6% as freezing drizzle and 1.8% as freezing rain. Only 1.3% of the precipitation was liquid, either drizzle or rain.

Fig.5: I am surprised that the PDF of SW/LW/Netsurface radiation balance do not differ much for the ship- vs. ice period. – Can you explain why?

SW: solar radiation is quite low so far north and albedo does not differ too much between transit and ice drift as transit was through heavy ice anyway. no open water periods are included in the statistics, so a similar radiation balance is really not surprising.

Fig.6: What do the grey stripes in d) represent? –Ok, seems like the answer is on p.10: No radar observations (and thus Cloudnet target classifications) were possible during the ice-breaking period between Aug 4-12. This should be mentioned much earlier. Also in Fig. 14 it should be explicitly stated in the caption that P2 had to be excluded.

We added the data availability to section 3 (see also answer above) and also to the captions of Fig. 6 (now Fig. 7) and Fig. 14 (Fig. 15).

**Reply to Reviewer 2**

- Introduction: Nicely done. The introduction properly motivates the AO2018 deployment within the broader context of Arctic research issues that still need to be addressed by the community. It is succinct, yet manages to provide ample background information. Reviews too often focus on items that should be changed or clarified, so I wanted to take the opportunity to express positive feedback regarding the introduction.

  Thank you for the positive feedback on the introduction.

- Lines 164-165: How is shortwave albedo quantitatively estimated using imagery?

  It is estimated by manually examining the imagery to estimate ice/water fractions and assigning an albedo value based on the area weighted average of typical albedos for ice and water.

- Line 168 and Fig. 4: How is near-surface temperature defined? Is this 2-meter temperature, or some other level?

  Shown and discussed are measurements undertaken on board the ship on the 7th deck (approx.. 25 m amsl). We added following explanation to the text:

  *Measurements of near-surface conditions undertaken on board the ship on the 7th deck (approximately 25 m above amsl) and at the foremast are shown in Fig. 5*

- Sections 4.3 and 4.4: A few sentences could benefit from simple restructuring or key comma insertions to reduce run-on sentences. Similar minor issues also appeared in other sections, but can also be rectified during final editing processes. I wanted to mention them here, though, since I found them marginally distracting. A few examples are provided below.
  o Lines 232-233: "If no temperature inversion could be identified, the strongest stable layer:"
  o Lines 239-240: ": to the lowest measurement heights of the radiosonde (30m), it was classified: : :"

  Thank you for the comment, we changed the sentences and went over the script, trying to improve readability in other sections as well.

- Lines 257-258: A general question regarding Cloudnet and radars employed in this campaign. The authors mention that Cloudnet does not observe 49% of fog events since the first usable radar observation is about 150m above ground level (i.e., many fog events are very shallow). Does radar sensitivity also affect fog detection statistics? Stated another way, it would be nice for the authors to advertise instrument sensitivity somewhere in the manuscript (e.g., minimum radar detectable signal).

  No radar sensitivity doesn't affect fog statistics as fog occurrence statistics were gained from the visibility sensor.

- General Cloudnet question: I assume Cloudnet algorithms are only applied to scanning radar observations and not the Micro Rain Radar (MRR)? Depending on how the MRR is configured, it can provide valuable observations below 150 m. But MRR sensitivity probably will not detect fog and non-precipitating clouds.

  Cloudnet is only applied to the cloud radar as non-precipitating clouds are not detected with the MRR.

- Section 4.4: How are cloud phases determined? Combined radar-lidar observations? Radar only?

  Cloud phase is determined using a combined radar-lidar-radiosonde approach. Liquid clouds are defined by a high lidar backscatter and a distinct decrease after the signal. Ice is identified by radar with downward pointing vertical velocity and a dew point temperature < 0°C. The melting layer can also be identified by the radar LDR > -15dB. This is all documented in the cloudnet papers & user documentation, referenced, and the literature upon which Cloudnet is based. In our opinion a detailed discussion is out of place in this paper.

- Lines 332-334 and a few other locations throughout the manuscript: I suggest economizing wording and removing nominalisations to improve readability. I definitely do not want to completely alter the authors' voice, but simple changes like the following will be impactful: "Equivalent potential temperature profiles are strongly stratified in the lowest 150 m, : : :."

  We rephrased the sentence and tried to improve other sentences as well. We hope readability has improved now.

- Section 6: This is an important section. I found myself begging for comparisons to previous research when results were presented in earlier sections. In hindsight, though, I like how the authors refrained from comparing to previous studies until this discussion like section. It serves as a nice overall summary that effectively complements the conclusions.

  Thank you for the comment.

- Conclusion: I like how the last paragraph begins, but I somehow feel that it ends in a disappointing fashion. It feels: : :.incomplete? I am not sure how to properly describe it. Maybe ending with a strong statement about how observations can help answer the lingering question of increasing Arctic cyclone activity shown by reanalyses? Simply swapping the second and third sentences might help, with an appropriate bridge that connects the first and second sentences. Something like "For instance, reanalysis data indicates an increase: : :: : :". Then relate the observations from the current study and need for continued observations to definitively answer this outstanding scientific question. The manuscript will not suffer tremendously if the last paragraph is not altered, but I encourage the authors to concoct a more impactful ending paragraph.

  We have followed this advice and changed the conclusion. We hope it is now more satisfying. This is the new conclusion paragraph:

  *Overall, the meteorological results from AO2018 summarised here provide a guide for further investigation. For instance, reanalysis data already shows an increase of Arctic cyclone activity during the second half of the twentieth century (Zhang et al., 2004) and global and regional climate models suggest a further increase of cyclone activity during summer over the Central Arctic by the end of the 21$^{st}$ century (Orsolini and Sorteberg, 2009; Nishii et al., 2015; Akperov et al., 2019). This study shows that strong cyclonic activity is associated with changes of the thermodynamic structure, the cloud types and the vertical cloud distribution compared to previous results. It raises the question of whether this was an exceptional year or if these changes are representative of climatological change in Arctic summer atmospheric conditions.*

**Figure Comments**
- Fig. 2: Suggest adding [hPA] units to colorbars or figure caption.
    Added to caption

- Fig. 3: Wind speed colorbar labels quite difficult to discern. Are the blue and red numbers located at regular grid intervals the surface temperature?
    We increased the font size of the colorbar labels. Yes, numbers are surface temp

- Fig. 4: Suggest adding "vertical dashed lines" to the last sentence of the caption to distinguish between horizontal dash-dot lines in a few of the figure panels. Also consider adding "(e.g., P1, P2, : : :, P8)" to the figure caption to explicitly advertise that these labels are associated with respective observational periods.
    Done

General Question: Is there any reason why precipitation statistics were not shown? I completely understand the need to draw a proverbial analysis line somewhere – every detail cannot be shown. But I am curious if precipitation statistics have been analysed or plan to be analysed in a separate study. At the very least, this topic could be added to the future research discussion in the conclusion.
    We added a little bit more detail on precipitation to the manuscript. We have replaced the precipitation intensity in Fig.4 with the accumulated precipitation amount. Furthermore, we added form of precipitation to Fig. 4 and to the discussion. The measurements show that almost all precipitation fell as snow or ice (93.3%), another 4.6% as freezing drizzle and 1.8% as freezing rain. Only 1.3% of the precipitation was liquid, either drizzle or rain.
    However, as already stated from the reviewer, there are limits on how much we can add to the manuscript. Further analysis is planned.

- General Question: Did this campaign encounter any specific measurement complications (e.g., instrument performance, logistics, etc.)? If yes, it would be great to briefly describe some of them to both advertise how difficult it is to operate shipborne instrumentation thousands of kilometers from population centers in a rather hostile environment and serve as valuable feedback for other researches that might consider adopting similar measurement techniques in future field campaigns.
    No, there were no significant instrumentation issues worth documenting in the manuscript. The remote sensing instruments in general have been found to operate very well in spite of the harsh Arctic conditions.